# Structural organization of HBV pgRNA genome driven by phase separation in capsid confinement

Yunqiang Bian [1,2,5], Hai Pan [1,5], Jiaqi Mao[2,3,5], Yixin He[1,2], Yanwei Wang[3], Yi Cao [2], Wenfei Li [2,4] ✉ & Wei Wang [2]

Viruses rely on the precise packaging of their genomes within a capsid to execute essential life-cycle events, yet the principles governing genome structural organization in this confined environment remain elusive. Here, we reveal that hepatitis B virus (HBV) pregenomic RNA (pgRNA) exploits liquid-liquid phase separation (LLPS) inside the capsid to sculpt its architecture. Multiscale molecular dynamics (MD) simulations, supplemented by biochemical assays, show that pgRNA coalesces into a hollow, shell-like condensate along the inner capsid surface, with coexisting low- and high-density regions. Electrostatic interactions between pgRNA and the disordered C-terminal domain of capsid protein primarily govern condensate formation. LLPS drives the establishment of microphases composed of nematically aligned RNA hairpin arrays interspersed by domains rich in flexible single-stranded RNA linkers, achieving an optimal balance between structural order and dynamic flexibility. Intriguingly, although the ensemble-averaged pgRNA density exhibits icosahedral symmetry, individual simulation snapshots display pronounced heterogeneity, indicating symmetry breaking at the single-particle level. In addition, LLPS-induced hollow-shell architecture of pgRNA genome promotes long-range RNA base-pairing and enhances polymerase mobility, which may facilitate the functional dynamics of polymerase during reverse transcription. Our findings uncover a capsid-confined LLPS mechanism that orchestrates viral genome structure and dynamics, offering new targets for antiviral intervention.

Viruses are nanoscale pathogens with their genetic material encapsulated in a protein capsid. Due to the polyanionic nature of nucleic acids and the restricted volume of the capsid, the viral genome is often tightly compacted with significant electrostatic repulsion[1–3]. Some viruses even execute key biochemical events, e.g., genome replication, within this densely crowded environment[4,5]. Consequently, a fundamental question arises: how is the viral genome structurally organized to fit the confined capsid space while still enabling essential biochemical functions? Addressing this question holds profound significance, because it not only advances our understanding of the key molecular mechanisms driving viral life cycle but also provides critical insights for designing targeted vaccines and antiviral therapies[6–9].

[1]Wenzhou Key Laboratory of Biophysics, Wenzhou Institute, University of Chinese Academy of Sciences, Wenzhou, Zhejiang, PR China. [2]Department of Physics, National Laboratory of Solid State Microstructure, Nanjing University, Nanjing, PR China. [3]Department of Physics, Wenzhou University, Wenzhou, PR China. [4]Jiangsu Key Laboratory for Cardiovascular Information and Health Engineering Medicine, Department of Cardiology, Nanjing Drum Tower Hospital, Medical School, Nanjing University, Nanjing, PR China. [5]These authors contributed equally: Yunqiang Bian, Hai Pan, Jiaqi Mao. ✉e-mail: wfli@nju.edu.cn

The structural organization of viral genomes has long been a focus in structural biology research[10–15]. Breakthroughs in cryo-electron microscopy (cryo-EM) have enabled the determination of viral genome architectures at subnanometer resolution for several viruses, although the atomic resolved structural determination of viral genome remains unfeasible[11,16]. Complementing these experimental advances, molecular dynamics (MD) simulations have emerged as a powerful tool for probing the dynamic processes underlying viral genome packaging, offering insights that are otherwise difficult to capture experimentally[17–22]. For instance, Coshic et al. performed multiscale MD simulations for the bacteriophage HK97, which is a pressurized double-stranded DNA (dsDNA) virus with its genome compactly packaged within a preformed capsid under the assistance of packaging motor[19]. They uncovered a loop-extrusion mechanism of the motor-powered dsDNA encapsulation and produced all-atom structural models for the entire virion. The genomes of the densely packed dsDNA viruses often show clear order in structures within their capsids[23,24]. In contrast, for some viruses containing single-stranded RNA (ssRNA), e.g., the pregenomic RNA (pgRNA) intermediate of Hepatitis B virus (HBV) and the majority of icosahedral ssRNA viruses, their genomes can exhibit high conformational flexibility[10]. This inherent dynamics complicates traditional structural characterization methods, leaving the organizational principles of these viral genomes poorly understood. In this study, we investigate the structural organization and dynamics of HBV pgRNA genome within the viral capsid through multiscale MD simulations and experimental validation.

HBV is an icosahedral virus with its capsid composed of 240 capsid core protein (Cp) monomers arranged in T = 4 symmetry (Fig. 1a). The Cp comprises two domains: a well-structured N-terminal domain (NTD; residues 1–149) and an intrinsically disordered C-terminal domain (CTD; residues 150–185) (Fig. 1b). It is important to note that HBV serotypes may exhibit distinct lengths (e.g., 183 aa). The NTDs assemble into the outer capsid shell, while the unstructured CTDs, which are enriched in arginine residues (termed as R-arm or ARM), shape up a positively charged inner capsid surface essential for genome packaging[25,26]. Although HBV is a dsDNA virus, its replication cycle involves a pregenomic RNA intermediate. During early stage of replication, viral pgRNA and polymerase co-assemble with capsid proteins to form immature virions[27]. DNA synthesis takes place in capsid via reverse transcription with pgRNA functioning as the template, during which the dynamic remodeling of pgRNA conformations plays a crucial role. Along with the DNA synthesis, pgRNA is hydrolyzed by the ribonuclease domain of the polymerase, which is then followed by synthesis of the complementary DNA strand to produce mature HBV. The structural organization and dynamics of pgRNA in capsid is central to its role in reverse transcription, making it a prime target for antiviral therapies[28]. Cryo-EM revealed an icosahedrally ordered density distribution of the encapsulated pgRNA with the CTDs acting as a nucleic acid chaperone[29]. In addition, MD simulations have elucidated asymmetric fluctuations of capsid subunits and ion transport dynamics across the capsid[18,30]. However, how pgRNA is structurally organized to enable its functional dynamics within capsid remains elusive.

To address this question, we investigate the structural organization and dynamics of HBV pgRNA within the capsid combining all-atom MD simulations, coarse-grained (CG) MD simulations, and biochemical experiments. Strikingly, the results demonstrate that the organization of the pgRNA genome is governed by liquid-liquid phase separation (LLPS) mediated protein-RNA co-condensation. The electrostatic interactions are the dominant driving force of the co-condensation between the unstructured CTDs and the pgRNA chain along the inner surface of the capsid. These CTD-pgRNA condensates exhibit liquid-like dynamics, with the pgRNA retaining high mobility. The viral polymerase tends to partition out of the condensate and into the dilute regions of the capsid interior, enabling rapid diffusion, which is a prerequisite for efficient reverse transcription. Intriguingly, while the density distributions of the CTD-pgRNA condensates in the individual snapshots sampled by MD simulations are highly heterogeneous, the ensemble-averaged density distribution recapitulates the icosahedral symmetry patterns as observed in cryo-EM maps, which suggests symmetry breaking of pgRNA organization in the individual virions. Notably, CTD-pgRNA condensates formed by LLPS exhibit microphase structure featured by nematically aligned RNA hairpin arrays interspersed by flexible single-stranded RNA linkers, achieving an optimal balance between structural order and dynamic flexibility. This study provides the first instance of a virus harnessing phase separation to orchestrate genome organization within a confined capsid space, though phase separation is well-documented in wide range of host-cell processes like ssRNA-mediated capsid assembly[31–34]. Targeting the phase behavior of CTD-pgRNA condensates could thus represent a novel antiviral strategy.

## Results

### Structural organization of pgRNA genome in HBV capsid

We first performed all-atom MD simulations with explicit solvent to elucidate the global structural characteristics of pgRNA within the capsid. The missing CTDs in the PDB structure of HBV capsid (PDB code: 3J2V) were reconstructed to generate a complete capsid. In the initial configuration of the MD simulations, the pgRNA with the wide-type sequence was placed inside the capsid in a compact conformation without pre-assigned secondary structures (Fig. 1c and Methods). During the simulations, pgRNA quickly condensed along the inner surface of the capsid, facilitating the emergence of a hollow conformation (Fig. 1d), which is in close agreement with the cryo-EM data observed in previous experimental studies[29]. The radial distribution of atomic density offers a direct demonstration of this structural feature (Fig. 1e). Notably, the density distributions of the CTDs and pgRNA are largely overlapping, indicating their co-condensation driven by CTD-pgRNA interactions. Furthermore, the density distribution within this spherical shell is non-uniform, with the coexistence of low and high density regions (Fig. 1f and Supplementary Movie 1).

To further assess the dependence of the results on the initial structure and force fields, we conducted two additional sets of all-atom MD simulations initiated from pgRNA structures without pre-assigned secondary structures and with predicted secondary structures, respectively. Both simulations employed an alternative force field (amber14sb_OL15 force field with NBFIX correction)[35,36]. Consistent with the above result (Fig. 1d), these simulations also revealed rapid condensation of pgRNA along the inner surface of the capsid, forming a shell-like arrangement (Supplementary Fig. 1a–d and Supplementary Movies 2, 3). These results indicate that the observed pgRNA condensation is a robust process, largely insensitive to the choice of initial pgRNA structures and force fields.

By analyzing ion concentrations from the all-atom MD simulations, we found that the number of mobile ions flowing in and out of the capsid becomes comparable within 10 ns (Supplementary Fig. 2a). Moreover, time-dependent radial distributions of ion concentrations converge within the simulation timescale (Supplementary Fig. 2b, c). These results suggest that the Donnan equilibrium can be established within the timescale of the all-atom MD simulations. The converged radial distributions further demonstrate that the distribution of mobile ions inside the HBV capsid exhibits notable heterogeneity (Supplementary Fig. 2d, e). In addition, the average concentration of $Na^+$ inside and outside the capsid are 0.20 M and 0.086 M, respectively. The resulted relative ratio of the $Na^+$ concentrations between the interior and exterior of the capsid (~2.3) is considerably smaller than the values reported for other dsDNA virus (>10)[19]. A likely explanation for this weaker Donnan equilibrium effect is that only a single RNA strand is present at the pgRNA intermediate stage of the HBV life cycle. In this case, the positive charges of the CTD segments (+3600 in total) and the negative charges of pgRNA (−3200) are approximately balanced.

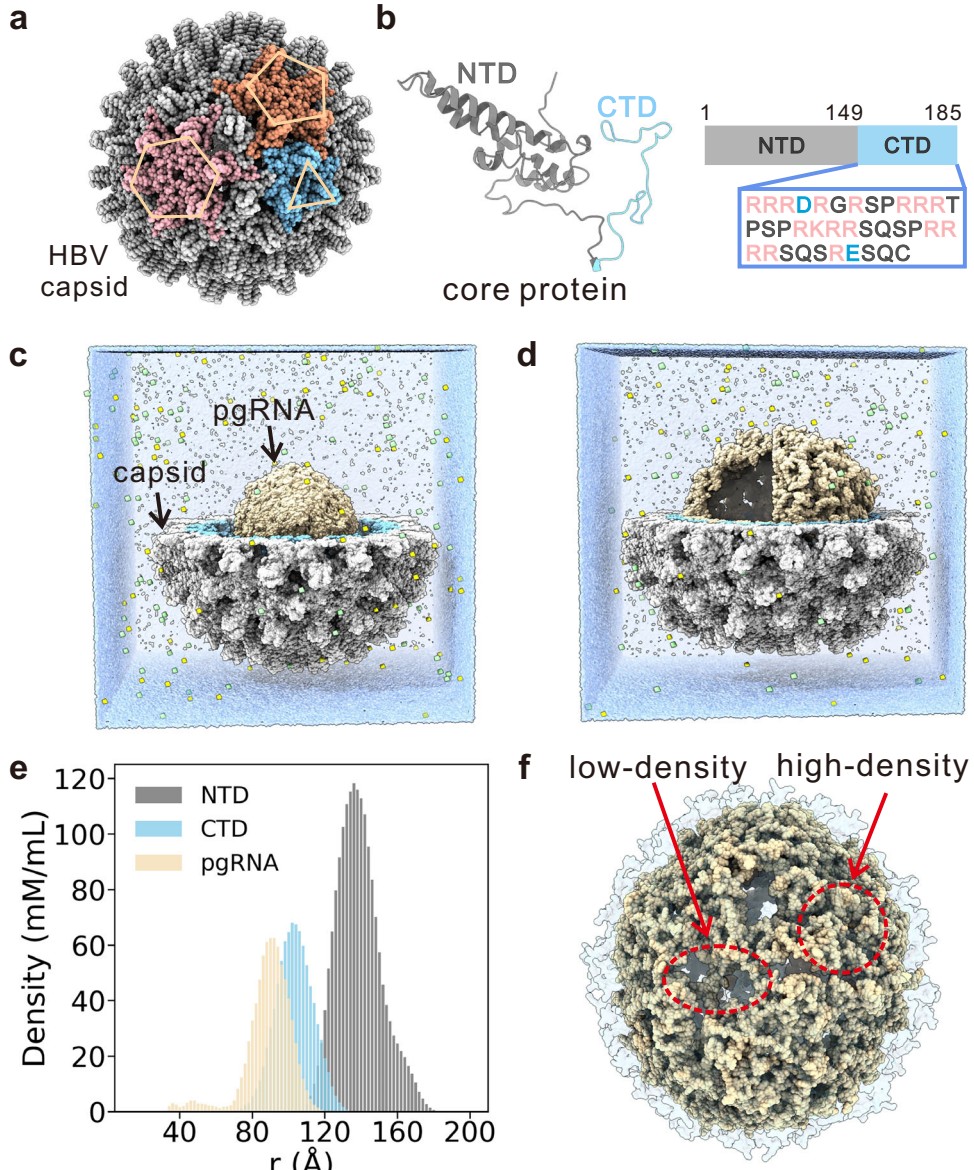

**Fig. 1 | All-atom simulation of the pgRNA-encapsulated capsid of HBV.**
**a** Structure of the HBV capsid. Representative threefold, fivefold and quasi-sixfold symmetry sites are colored blue, orange and pink, respectively. **b** Three-dimensional structure of HBV capsid protein. The NTD and CTD are colored gray and blue, respectively. The sequence of the CTD is also displayed, with positively charged residues in red and negatively charged residues in blue. **c** Surface representation of the initial configuration for MD simulations. Only half of the capsid is shown for clarity. The pgRNA is colored wheat. Na$^+$ and Cl$^-$ ions are represented by yellow and green spheres, respectively. **d** Surface representation of the final snapshot of the MD simulation. Half of the capsid is shown and octant of pgRNA is removed for clarity. **e** Radial distribution functions of NTD (gray), CTD (blue) and pgRNA (wheat) relative to the capsid center, highlighting their spatial organization. **f** Representative conformation of CTD-pgRNA condensates depicting the coexisting low- and high-density regions, with CTDs shown in a transparent representation. Source data are provided as a Source data file.

With the progression of reverse transcription of pgRNA into dsDNA, the negative charges of the genome will significantly exceed the positive charges of CTDs, which would lead to more pronounced Donnan equilibrium effect.

It is intriguing to further investigate the biophysical mechanism driving the formation of the above observed structural organization of pgRNA genome. Physically, the observed coexistence of low- and high-density regions is often a hallmark of phase separation. Notably, LLPS of biomolecules has been widely implicated in many biological processes, such as chromatin organization, stress responses, and DNA damage repair[37–46]. Aberrant condensation is linked to the pathogenesis of cancer and numerous neurodegenerative diseases. To elucidate that the above observed structural organization of pgRNA within the capsid is an outcome of phase separation, we further investigated its structure and dynamics in response to typical environmental modulators of LLPS, including salt concentration ($C_{salt}$) and temperature (T) in the subsequent sections.

**Phase separation of pgRNA genome within capsid confinement**
Because all-atom MD simulations of the phase behaviors of biomolecules is computationally challenging, CG MD simulations were utilized to explore the phase behavior of pgRNA by manipulating the salt concentration $C_{salt}$ and temperature $T$. In the CG simulations, each residue is represented by one spherical bead located at the $C_\alpha$ position (Fig. 2a), and each nucleotide is represented by three beads corresponding to the phosphate group (P), sugar ring (S), and base (B), respectively (Fig. 2b). The force fields are described by the AICG2+ model and 3SPN.2 model[47–49], which have been successfully used in

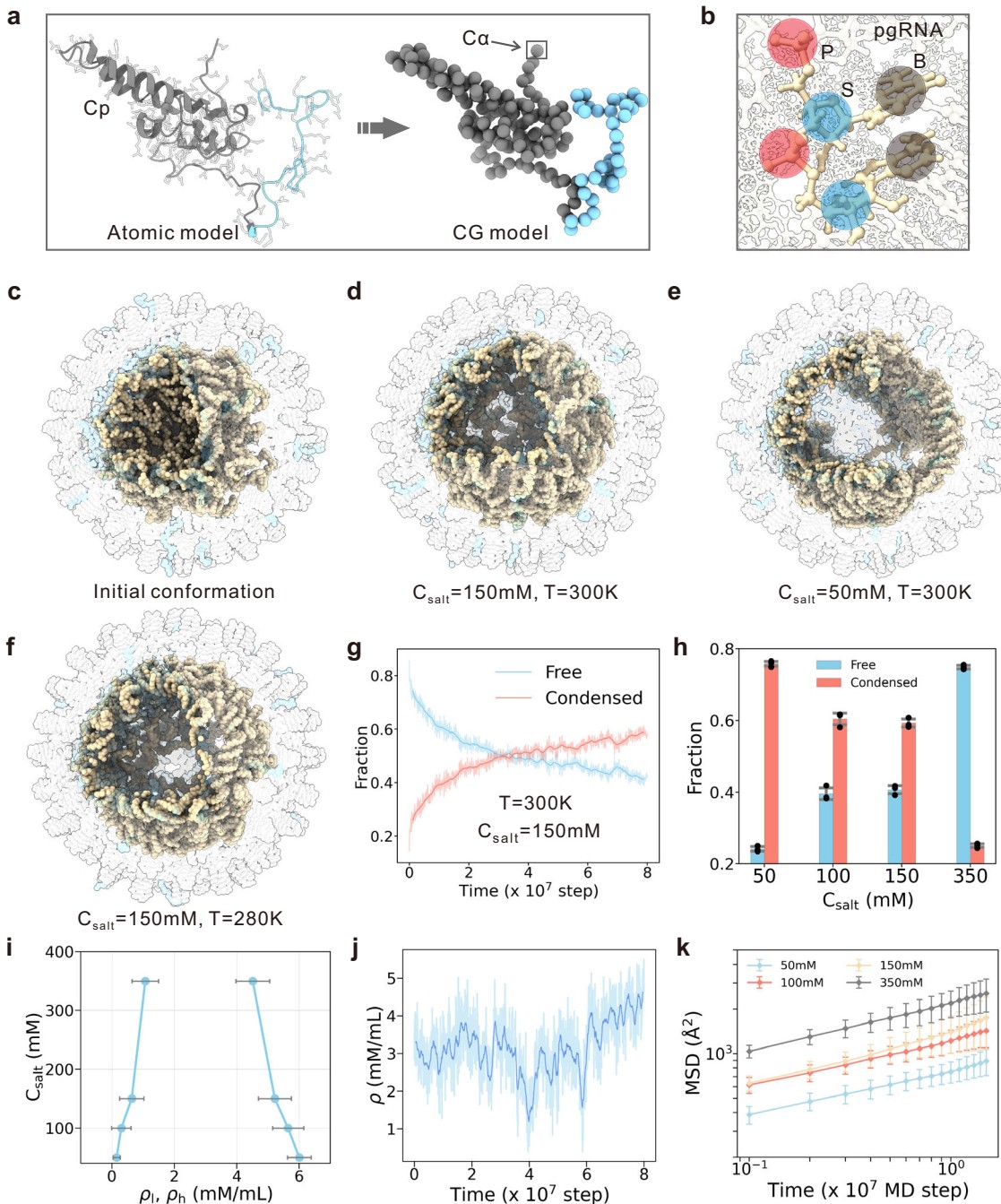

**Fig. 2 | Coarse-grained simulations of pgRNA phase separation modulated by ionic strength and temperature.** Schematic illustration of the CG models of core protein (**a**) and pgRNA (**b**). The NTD and CTD of core protein in (**a**) are corlored gray and blue, respectively. The CG phosphate, sugar and base groups in (**b**) are colored red, blue and gray, respectively. **c** Initial structure in the CG simulations (octant removed for clarity). The NTDs, CTDs and pgRNA are depicted in gray, blue and wheat, respectively. **d**–**f** Representative structures sampled by CG simulations under varying salt concentrations and temperatures. The same color scheme in panel (**c**) was applied. **g** Fraction of free (blue) and condensed (red) nucleotides as a function of time. Nucleotide acids are categorized into free and condensed phases if they belong to a small (≤4 nucleotides) and large (>4 nucleotides) clusters, respectively. The shaded line and the solid line denote the raw data and the averaged data, respectively. **h** Fraction of free (blue) and condensed (red) nucleotides as a function of salt concentrations ($C_{salt}$). Data are presented as mean +/− standard deviation (SD) derived from 3 independent simulations. **i** Phase diagram as a function of salt concentration ($C_{salt}$), mapped via the mean densities of the low- ($\rho_l$) and high-density ($\rho_h$) regions. Data are presented as mean +/− SD derived from 1000 independent simulation snapshots under each salt concentration. **j** Temporal course of local RNA-bead density within a fixed bin at a fivefold capsid site. The same plotting scheme in panel (**g**) was applied. **k** Mean-squared displacement (MSD) of a representative pgRNA segment (nucleotides 1–50) under different salt conditions. Data are presented as mean +/− SD derived from 32 RNA segments of 100-nt in length. Source data are provided as a Source data file.

modeling the structural dynamics of protein-nucleic acid complexes[50,51], such as the nucleosome assembly and remodeling[52,53], the protein binding induced DNA bending[54], the dynamics of parental H3/H4 recycling at the replication fork[55], and the LLPS of protein-ssDNA mixture[56]. More details of the CG model are given in Methods.

Initiating from a uniformly distributed random conformation of pgRNA in the capsid (Fig. 2c), we performed three independent MD

simulations across varying $C_{salt}$ and T. In all simulations, pgRNA rapidly condensed into a spherical shell colocalized with CTDs. We also conducted CG simulations initiated from pgRNA structures with predicted secondary structures, and comparable results were observed (Supplementary Fig. 1e, f and Supplementary Movie 4). Under physiological conditions, concurrent segregation of pgRNA and CTDs into coexisting low- and high-density regions along this spherical shell was clearly observed (Fig. 2d), which is consistent with the results of all-atom MD simulations. CG simulations under different conditions further revealed that phase separation extent is sensitive to $C_{salt}$ or T (Fig. 2d–f and Supplementary Figs. 3, 4a). At lower $C_{salt}$ or T, pgRNA exhibited more pronounced phase separation, forming grid-like morphology with heterogeneous density along the spherical shell, whereas an increment in $C_{salt}$ or T leads to a more uniform morphology.

To characterize phase separation behavior, we categorized nucleotides into "free" and "condensed", depending on whether they form large clusters. As shown in a representative MD simulation trajectory under physiological conditions (Fig. 2g), the nucleotides in the free state were observed to progressively coalesce into condensed state. By the end of the simulation trajectory, more than half of the nucleotides coalesced into condensates, while a substantial fraction remained free. Concurrently, low-density pores emerged throughout the spherical shell. These observations clearly illustrate a phase separation that results in the formation of substructures with different densities. Furthermore, under conditions of low salt concentration and temperature (Fig. 2h and Supplementary Fig. 4b), the nucleotides exhibited a more pronounced phase separation with increased content of condensates, consistent with the above conformational analysis. We then quantified phase behavior by dividing the pgRNA spherical shell into discrete bins and computing pgRNA densities within each bin (Methods). Phase diagram displays the coexistence of low- and high-densities ($\rho_l$ and $\rho_h$) across wide range of $C_{salt}$ and T (Fig. 2i and Supplementary Fig. 4c). This bimodal density distribution and its environmental dependence support again the phase separation of pgRNA genome.

Given that biomolecular condensates formed via LLPS typically exhibit liquid-like properties, we further investigated the fluidity of the encapsidated pgRNA genome. By calculating the density of RNA nucleotides within a bin located at a fivefold symmetry site over time (Fig. 2j), we observed significant fluctuations, indicating that the pgRNA genome is highly dynamic and the nucleotides are exchangeable between low- and high-density regions. By dividing the pgRNA sequence into 100-nucleotide segments, we also calculated the mean square displacement (MSD) averaged among these segments from the aforementioned CG MD simulations. The MSD results reveal substantial diffusivity (Fig. 2k and Supplementary Fig. 4d), particularly under physiological conditions (Supplementary Movie 5). At longer lag-time scale, the slope of the MSD plot decreases due to confinement within the capsid (Supplementary Fig. 5). These observations suggest that the pgRNA maintains a high degree of fluidity despite its association with the inner surface of the capsid, supporting a liquid-like condensate organization, rather than a fully ordered self-assemble structure under physiological conditions.

### Phase separation of pgRNA genome in bulk system

To directly assess the LLPS propensity of CTD-RNA mixtures, we carried out slab coexistence simulations, which is a benchmark method for characterizing biomolecular phase behavior[57–59]. The simulation system contains 216 protein chains corresponding to the disordered CTD segments and 216 15-mer polyadenosine oligonucleotides ($A_{15}$), recapitulating the protein-RNA stoichiometry found in pgRNA-encapsulated HBV capsids. By this setup, the geometric restraints arising from capsid anchors and contiguous RNA chain are eliminated, and the simulation results will be dominated by the intrinsic physicochemical properties of the two components. At 300 K and 150 mM salt,

the z-axis density profiles reveal two coexisting phases: a dense condensate enriched in inter-chain CTD-RNA contacts, and a dilute phase of freely diffusing molecules (Fig. 3a, b and Supplementary Fig. 6; Supplementary Movie 6). This bimodal demixing persists across a wide range of salt concentrations, as shown by the distinct bimodal boundary in the phase diagram (Fig. 3c). These findings clearly establish that the CTD-RNA system is capable of undergoing phase separation to yield coexisting dense and dilute phases.

We then experimentally validated phase separation in CTD-RNA solutions composed of CTD peptides and $A_{15}$ RNA chains. At 10 µM concentration for each of the two components, we observed characteristic LLPS features including droplet growth and spontaneous droplet fusion events by bright field light microscopy (Fig. 3d, e and Supplementary Movie 7), confirming liquid-like condensate formation. Further variation of salt concentration revealed a strong dependence of phase separation on electrostatic interactions (Fig. 3f). Condensates formed readily at low and moderate salt concentrations but were completely suppressed at 500 mM salt, demonstrating that the electrostatic interactions between CTDs and RNA molecules play a crucial role in driving condensation. To probe hydrophobic contributions, we added 1,6-hexanediol, a compound known to disrupt hydrophobic interactions, into the solution at the $C_{salt}$ of 150 mM. Phase separation persisted even at a high concentration of 1,6-hexanediol (20% v/v) (Fig. 3g), indicating that hydrophobic interactions are not the primary driving force behind phase separation.

We next assessed how CTD-RNA stoichiometry modulates LLPS by varying CTD concentration against a fixed 10 µM RNA background at 150 mM NaCl (Fig. 3h and Supplementary Fig. 7). Condensates formed over a broad CTD:RNA ratio. However, when the ratio dropped below 1:15, phase separation diminished, which may be attributed to the electrostatic repulsion from excess anionic RNA molecules. These observations further underscore that attractive electrostatic interactions between cationic CTDs and anionic RNA are the principal forces driving condensate formation.

### Surface-induced condensation of pgRNA genome

While LLPS-driven protein-RNA co-condensation has been extensively characterized in systems with freely diffusion components, HBV CTDs are tethered to the icosahedrally arranged capsid core. This suggests that the phase separation observed in our study represents a surface-induced CTD-pgRNA condensation. To investigate whether the observed coexistence of low- and high-density regions of pgRNA genome in capsid is related to the spatial heterogeneity of CTDs due to the icosahedral arrangement, we performed control simulations with CTDs uniformly arranged in a two-dimensional planar array. In this two-dimensional system, the NTD of the capsid protein was omitted, and the N-terminus of the peptides corresponding to CTDs were anchored to a two-dimensional plane by applying a harmonic potential. The reference positions were uniformly distributed at a density approximately equivalent to that of the CTDs in the native capsid. Molecular simulations across varying salt concentrations at 300K revealed phase behavior analogous to capsid-confined systems upon the addition of pgRNA (Fig. 4b–f). Under physiological conditions, clear phase separation emerged, manifesting as coexisting high- and low-density regions (Fig. 4e and Supplementary Movie 8). The phase separation became more pronounced at low $C_{salt}$ (e.g., 100 mM), while it was nearly indiscernible at high $C_{salt}$ (e.g., 350 mM). In addition, no phase separation was observed for the CTD-only system (Supplementary Fig. 8). Similar phase-separation behaviors were observed when much shorter RNA segments ($A_{15}$ RNA segments) were added (Supplementary Fig. 9). Interestingly, we observed clear coarsening events, in which two small clusters coalesced into a larger cluster (Supplementary Fig. 10 and Supplementary Movie 9), a hallmark feature of macroscopic phase separation. However, because one terminus of each CTD is anchored, the sizes of the high-density clusters remain

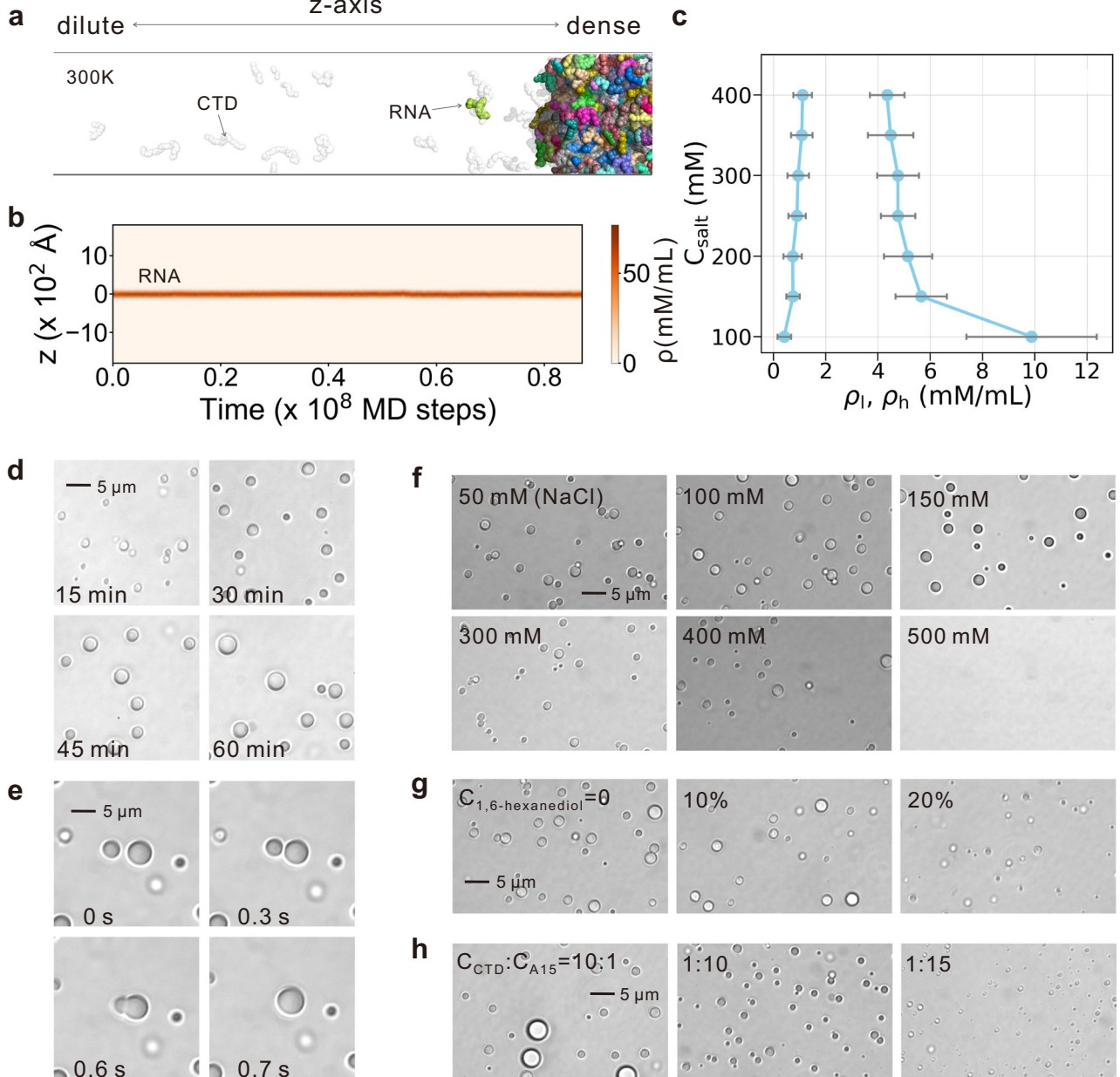

**Fig. 3 | Molecular simulations and biochemical assays for the phase separation of CTD-RNA mixture in bulk condition. a** Representative snapshot from slab simulations showing coexisting dense condensate and dilute phase. Only half of the conformation is shown for clarity. **b** Z-axis density profiles of RNA segments in the slab simulation trajectory. **c** Phase diagram from slab simulations with varying salt concentrations ($C_{salt}$), mapped via the mean densities of the dilute ($\rho_l$) and dense ($\rho_h$) phases. Data are represented as mean +/− SD derived from 1000 independent simulation snapshots under each salt concentration. Representative droplet growth (**d**) and fusion (**e**) events observed by bright field light microscopy at the salt concentration of 150 mM. Bright field droplet assays with varying salt concentrations (**f**), 1,6-hexanediol concentrations (**g**), and CTD:RNA molar mixing ratios (**h**). Source data are provided as a Source data file.

limited and continuous coalescence into larger droplets becomes difficult. These findings, corroborated with the corresponding phase diagram (Fig. 4b), imply that the structural organization of pgRNA genome driven by surface-induced phase separation is the intrinsic propensity of the CTD-pgRNA mixture system, which is independent of the icosahedral arrangement of capsid proteins.

To further examine the surface-induced condensation of pgRNA genome, we analyzed its density distribution along the two-dimensional surface. As expected, ensemble-averaged profiles from MD simulations at $C_{salt} = 150$ mM yielded a uniform pattern (Fig. 4g), reflecting global symmetry. In contrast, individual simulation snapshots revealed pronounced heterogeneity and asymmetry in pgRNA

organization, consistent with phase separation-driven patterning (Fig. 4h, i). This variability across conformations underscores the dynamic and liquid-like nature of pgRNA-CTD condensates under physiological conditions. Interestingly, the highly fluid feature of HBV-RNA interactions within the capsid has been observed by Taji et al through nuclear magnetic resonance (NMR) studies,[60] consistent with the liquid-like dynamics of the CTD-RNA condensate described in this work.

Structural analysis of the high- and low-density regions in the two-dimensional condensate layer reveals that, although one terminus of each CTD is uniformly grafted to a planar surface, the overall CTD density distribution is highly heterogeneous. Typically, the free ends of

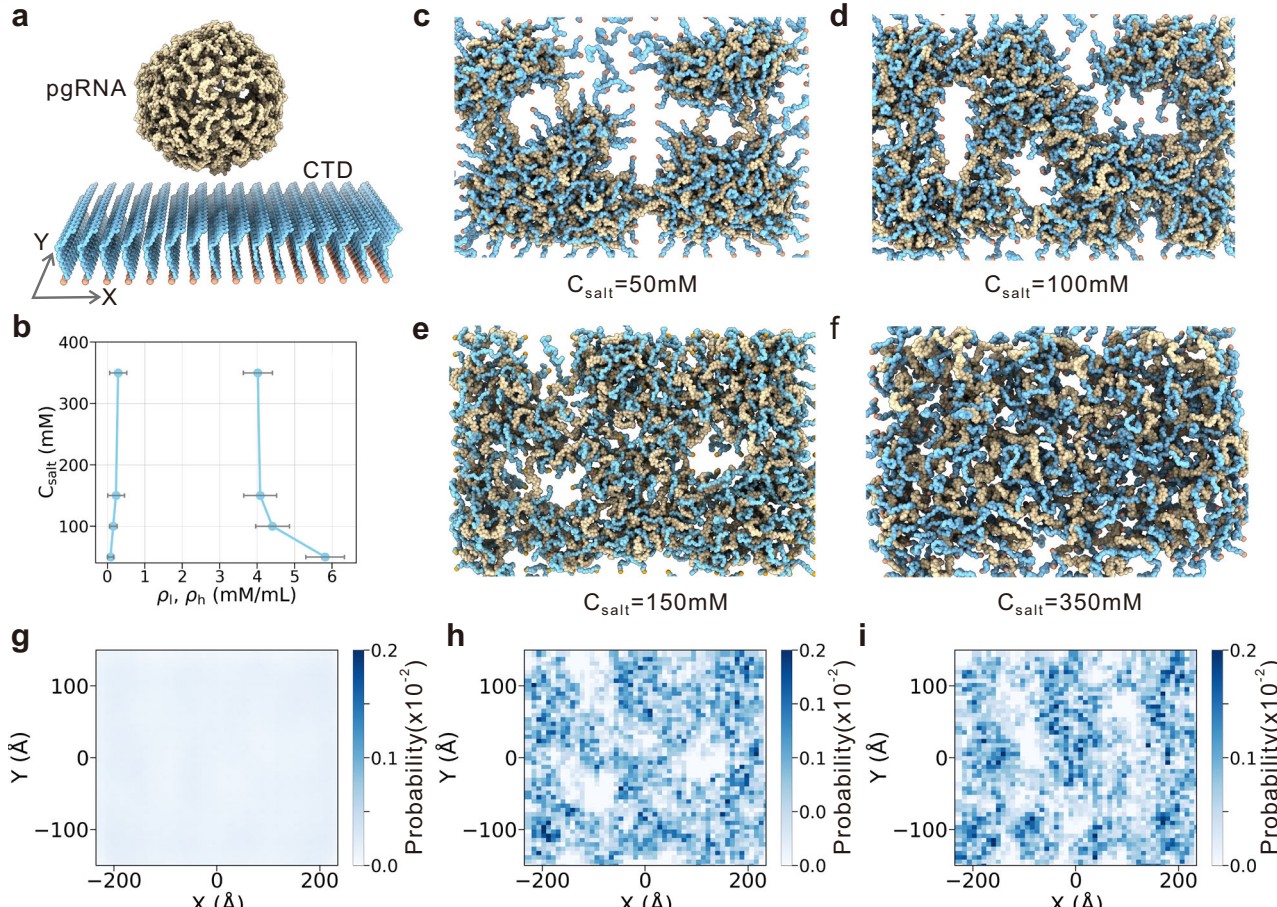

**Fig. 4 | Molecular simulations of pgRNA genome along two-dimensional CTD membrane. a** Initial structure in the CG simulations. The positions of the N-terminal residues (orange), which are uniformly arranged in a planar lattice (inter-particle distance, $d_N = 5$ Å), are fixed during MD simulations. **b** Phase diagram from slab simulations across varying salt concentrations ($C_{salt}$), mapped via the mean densities of the low- ($\rho_l$) and high-density ($\rho_h$) regions. Data are represented as mean +/− SD derived from 1000 independent simulation snapshots under each salt concentration. Representative simulation snapshots at salt concentrations of 50 mM (**c**), 100 mM (**d**), 150 mM (**e**), and 350 mM (**f**). **g** Ensemble-averaged density distribution along the two-dimensional membrane at salt concentration of 150 mM. **h, i** Density distributions along the two-dimensional membrane for two individual snapshots sampled from MD simulations at the salt concentration of 150 mM. Source data are provided as a Source data file.

CTD chains tend to orientate from low density regions toward high density regions (Supplementary Fig. 11a, b), leading to the formation of nanometer-sized clusters. Consequently, high-density regions are enriched in both CTD and RNA, whereas low-density regions contain relatively fewer CTD moieties due to covalent anchoring. In the high-density regions, close contacts between positively charged CTD segments and negatively charged RNA segments promote favorable electrostatic attractions. Meanwhile, the larger inter-CTD distances (due to outward orientation) in low-density regions reduce unfavorable electrostatic repulsion. Thus, RNA dewetting is facilitated by the enhanced electrostatic attraction in high-density regions and the reduced electrostatic repulsion in low-density regions. Further investigations established that the two-dimensional phase separation of the CTD-RNA mixture system depends critically on the surface density of CTDs. Altering CTD density by changing inter-chain distances under physiological conditions diminished phase separation (Supplementary Figs. 11, 12), likely due to alteration of the electrostatic balance.

The two-dimensional planar system described above is analogous to a polymer brush. Previous studies have showed that polyelectrolytes grafted onto a two-dimensional substrate can segregate into coexisting domains in response to electrostatic regulation[61–63], a phenomenon often referred to as "lateral phase separation"[64] or "planar phase separation"[65]. Compared to typical polyelectrolyte brush system, RNA in the HBV capsid is not only a participant in the phase behavior as an electrostatic regulator, but also a central component of the condensates. To further support the two-dimensional phase separation of the CTD-RNA system, we employed atomic force microscope (AFM) to characterize the morphological transition. Because of the short length of CTD peptides, the resulting condensate are typically nanometer in size, which is too small to be detected by fluorescence microscope or AFM. To overcome this limitation, we grafted polyethylene glycol (PEG) onto a planar substrate and decorated the free PEG terminus with CTD segments. This extended the effective polymer length, enabling the formation of larger clusters upon RNA addition. AFM measurements revealed that RNA addition promotes morphological heterogeneity in the two-dimensional system (Supplementary Fig. 13), resembling the lateral phase separation observed in early studies[61]. See Supplementary Material for more details. Because AFM has limited resolution and it is difficult to absolutely rule out alternative origins for the observed morphological transition beyond phase separation, we also carried out additional all-atom MD simulations of the two-dimensional CTD layer mixed with RNA segments. These all-atom MD simulations clearly demonstrated RNA dewetting (Supplementary Fig. 14a, b). Correspondingly, the distributions of Na$^+$ and Cl$^-$ around the condensate layer are highly heterogeneous (Supplementary Fig. 14c, d) and closely correlated with

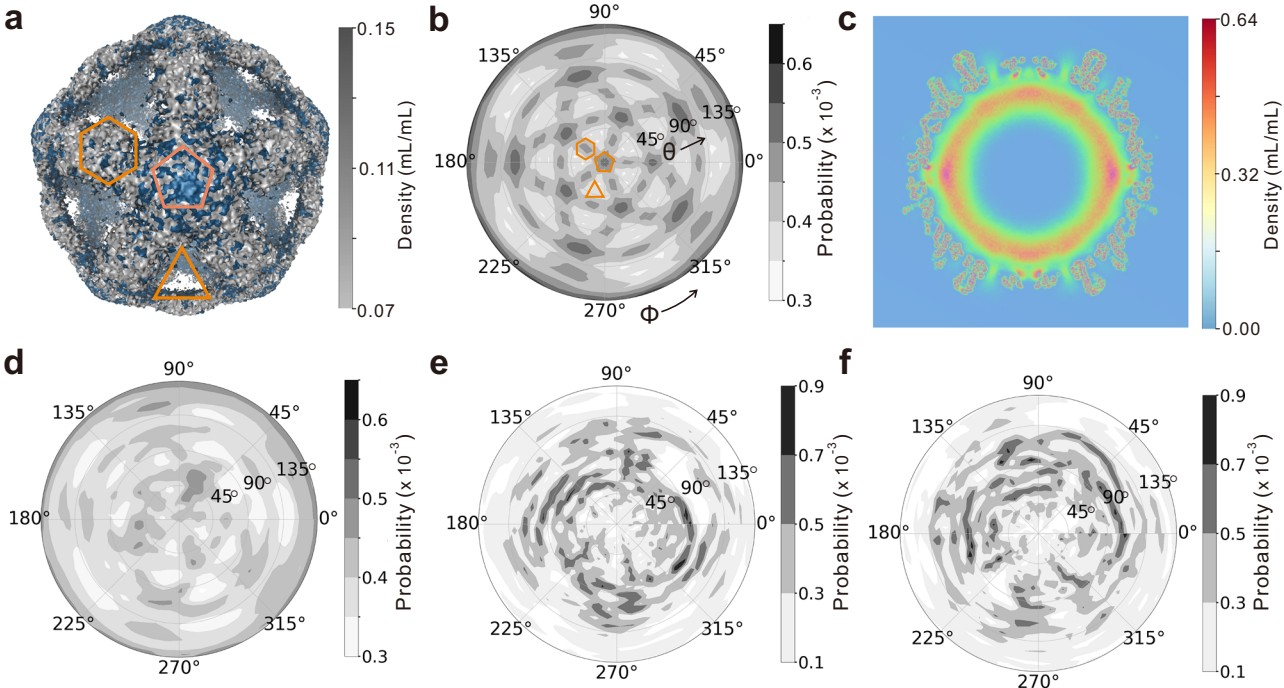

**Fig. 5 | Structural organization of pgRNA within capsid confinement from CG MD simulations. a** Ensemble-averaged density map of pgRNA (gray) and CTDs (blue) along the three-dimensional space within capsid. Rotational averaging was applied. The density less than the lower-limit of the density scale is not plotted. The threefold, fivefold and quasi-sixfold symmetry sites are labeled correspondingly. **b** Ensemble-averaged density distribution of pgRNA plotted along the two-dimensional spherical coordinates ($\theta$, $\phi$) with rotational averaging. The density profiles are calculated based on the pgRNA beads within a spherical shell (80 Å ≤ r ≤ 120 Å). **c** Volume slice plot of density distribution. **d** Ensemble-averaged density distribution of pgRNA plotted along the two-dimensional spherical coordinates ($\theta$, $\phi$) without applying rotational operation. **e**, **f** Density distributions of pgRNA plotted along the two-dimensional spherical coordinates ($\theta$, $\phi$) for two individual snapshots from MD simulations. Source data are provided as a Source data file.

the spatial organization of the CTD-RNA condensate (Supplementary Fig. 14e).

## Asymmetry in pgRNA genome of individual virion

Previous cryo-EM studies using multi-particle averaging revealed that the pgRNA genome adopts an icosahedrally symmetric conformation adapted to the HBV capsid geometry[29]. To examine whether our simulations recapitulate this characteristic symmetry observed in experiment, we computed three-dimensional density distributions of pgRNA and CTDs from the MD sampled conformational ensemble. Following the experimental protocol, we applied rotational averaging based on icosahedral symmetry[19]. The resulting density distributions for both CTDs and pgRNA components exhibit clear icosahedral symmetry (Fig. 5a), closely resembling prior cryo-EM data[29]. The icosahedral arrangement can be demonstrated more clearly by the projection of the pgRNA density distribution onto the two-dimensional spherical coordinates (Fig. 5b). Specifically, the populations are much enriched around the fivefold axis and quasi-sixfold vertices, while densities around the threefold axis are relatively lower. This high symmetry is also evident in the volume slice plot (Fig. 5c), consistent with experimental findings[29]. The enrichment of pgRNA around the fivefold and quasi-sixfold symmetry sites is attributed to the co-condensation of CTDs and pgRNA. As CTDs are relatively more concentrated at the fivefold and quasi-sixfold symmetry sites on the inner surface (Supplementary Fig. 15), the average density of pgRNA at these sites is consequently higher.

To assess whether the icosahedral density distributions are solely the results of rotational averaging, we also calculated the ensemble-averaged density distribution without applying rotational symmetry operations, using multiple independent MD simulations. One can see that the icosahedral symmetry is still evident, although it becomes much weaker (Fig. 5d). This indicates that the pgRNA genome organization is inherently modulated by the symmetric arrangement of capsid proteins. Notably, this pattern of reduced symmetry aligns with experimental reconstructions of HBV virions using asymmetric reconstruction methods[66]. Additionally, we analyzed the structural characteristics of pgRNA of the individual snapshots without ensemble-averaging. The pgRNA configurations from the individual snapshots of the MD simulations exhibit no discernible symmetry in their three-dimensional density profiles (Fig. 5e, f), suggesting that symmetry breaking occurs dynamically at the single-particle level despite global structural constraints.

## Base-pairing and microphase separation of pgRNA genome

The above combined MD simulations and experimental observations reveal that HBV exploits LLPS to regulate the spatial organization of its pgRNA genome within the capsid. To probe how LLPS influences pgRNA structural dynamics, we further analyzed the conformational features of pgRNA genome under LLPS regulation. A principal outcome of LLPS is the localized concentration of nucleotides, generating a microenvironment that favors secondary structure folding in pgRNA. Both all-atom and CG simulations demonstrate that base-pairing preferentially occurs in densely packed regions, facilitating the formation of dsRNA structures (Fig. 6a and Supplementary Fig. 16). Typical processes involving the formation of base-pairing interactions are presented in Supplementary Fig. 17 and Supplementary Movies 10–13. Representative CG trajectory (Fig. 6b) shows a marked increase in the number of base pairs ($N_{bp}$) coinciding with pgRNA condensation along the inner surface of HBV capsid, indicating that LLPS enhances pgRNA secondary structure. We also conducted additional CG simulations with the initial structure of the capsid-confined pgRNA possessing pre-assigned secondary structures, and similar role of LLPS in promoting

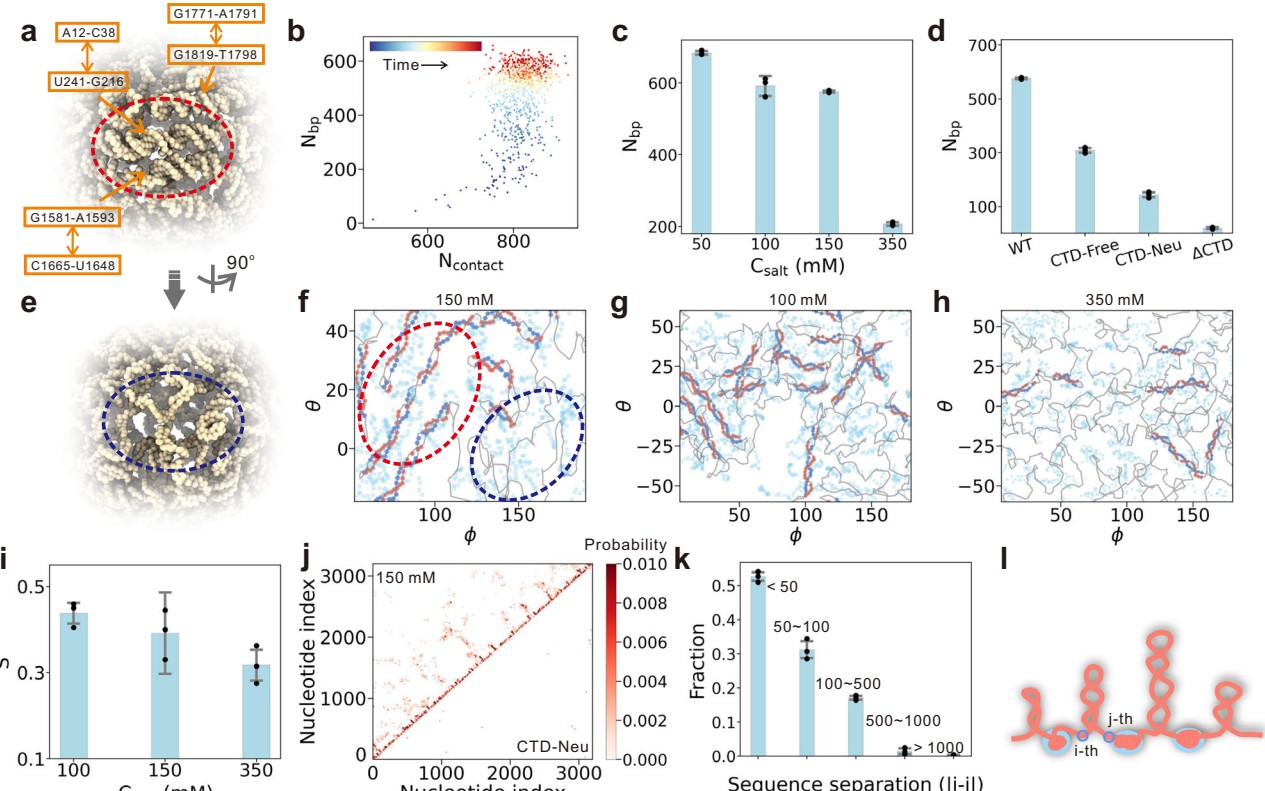

**Fig. 6 | Secondary-structure folding and microphase separation of pgRNA-CTD condensates. a** Representative CG simulation snapshot illustrating the formation of dsRNA structures of pgRNA. Complementary sequence segments involved in dsRNA formation are highlighted. **b** Correlation between pgRNA-CTD condensation, measured by the number of CTD-pgRNA contacts ($N_{contact}$), and pgRNA base pairing, measured by the number of base pairs ($N_{bp}$), along the time course of a representative MD trajectory. **c** Mean $N_{bp}$ as a function of salt concentration. **d** Comparison of mean $N_{bp}$ across different simulation systems. **e** Same as (**a**) but oriented to highlight ssRNA-rich region. **f**–**h** Projections of representative pgRNA/CTD structures onto two-dimensional spherical coordinates ($\theta$, $\phi$) at different salt concentrations to show the microphase separation. Gray traces denote pgRNA backbones; blue and red spheres mark dsRNA nucleotides; and light blue spheres represent CTD residues. **i** Scalar order parameter S of dsRNA pairs at differnt salt concentrations. **j** Contact map depicting base-pairing probability for the WT system (top) and CTD-Neu system (bottom). **k** Distribution of the sequence separation between paired dsRNA termini, revealing a predominance of local hairpin formation. **l** Schematic of the resulting tree-like architecture of CTD-pgRNA condensates, showing spatial segregation of ordered RNA hairpin domains and flexible ssRNA linkers. Data in (**c, d, i, k**) are presented as mean +/− SD derived from 3 independent simulations. Source data are provided as a Source data file.

base-pair formation was observed (Supplementary Fig. 18). Furthermore, $N_{bp}$ correlates with the extent of phase separation. Conditions promoting LLPS (i.e., low salt and low temperature) result in significantly elevated $N_{bp}$ values, whereas conditions that suppress LLPS reduce $N_{bp}$ accordingly (Fig. 6c and Supplementary Fig. 19). These results demonstrate again a mechanistic link between LLPS and pgRNA base-pairing within the HBV capsid.

The formed base-pairs exhibit pronounced dynamics in the CG simulations. Unwinding events of dsRNA can be clearly observed in the simulation trajectories (Supplementary Fig. 17b and Supplementary Movie 11). Moreover, substantial variation in the number of formed dsRNA segments during the CG MD simulations supports the highly dynamic nature of the dsRNA conformation (Supplementary Fig. 17e). The stability of dsRNA is length-dependent. Segments containing more base pairs exhibit longer lifetimes (Supplementary Fig. 17f), which were estimated using a maximum-likelihood method. While the mobility of the CTDs is constrained by their anchoring to the inner capsid surface, the encapsulated RNA displays considerable fluidity, as evidenced by dynamic changes in local density at specified capsid site (Fig. 2j) and by its significant mean squared displacement (Fig. 2k). The dsRNA segments also exhibit pronounced mobility, although markedly reduced compared to the ssRNA region (Supplementary Fig. 20e). This structural plasticity is functionally important for HBV pgRNA, as overly stable or rigid secondary structures could interfere with subsequent

reverse transcription. Furthermore, the formation of secondary structures through local base-paring in pgRNA, particularly sites involving packaging signals, has been proposed to play a critical role in the assembly of the HBV capsid.[67–70] Notably, Patel et al. demonstrated that folding of packaging sites into stem-loop structures promotes capsid assembly[67]. In line with this, our CG simulations showed that the segment corresponding to a potential packaging site (PS3) can spontaneously form stem-loop-like structure (Supplementary Fig. 17d). In addition, another important protein recognition motif, the cis-acting element $\varepsilon$, can also fold to stem-loop-like structure, although it involves frequent folding and unfolding transitions (Supplementary Fig. 17c).

To further substantiate the link between phase separation and pgRNA base-pairing, we conducted three control CG simulations designed to impair LLPS: (i) $\Delta$CTD, (ii) CTD-Neu, and (iii) CTD-Free (Fig. 6d). In the $\Delta$CTD simulation, CTD of capsid protein was truncated; in the CTD-Free simulation, the covalent linkage between CTD and NTD was severed, allowing CTD to diffuse freely; and in CTD-Neu, all charged residues of CTD were neutralized. As anticipated, all three conditions resulted in marked reductions in base-pair formation. Notably, the CTD-Free simulation retained co-condensation between CTD and pgRNA, although the condensates were diffusely distributed throughout the capsid interior and failed to adopt the characteristic spherical shell morphology (Supplementary Fig. 21). These findings

indicate that the CTD-induced alignment of pgRNA along the inner capsid surface is critical for stabilizing base-pairing interactions. In addition, both ΔCTD and CTD-Neu abolished phase separation, reinforcing the importance of electrostatic interactions in driving CTD-pgRNA co-condensation. Interestingly, the arginine-rich CTD has been suggested to act as a specific nucleic acid chaperone to facilitate the structural organization of the viral genome.[29,71] The CTD-induced pgRNA condensation and base-pairing observed in this study is in line with the chaperone role of CTDs as proposed in previous studies.

The detection of dsRNA within the HBV pgRNA genome indicates that its conformation is not fully disordered. Detailed structural analysis revealed a spatial segregation between dsRNA and ssRNA domains (Fig. 6a, e), with regions enriched in dsRNA distinctly separated from ssRNA-dominant areas (Fig. 6f), suggesting the emergence of microphase separation during pgRNA organization. This microphase separation was strongly modulated by ionic strength, with the segregation increasingly pronounced under lower salt conditions (Fig. 6f–h). Meanwhile, the dsRNA becomes shorter as the salt concentration increases (Supplementary Fig. 22a–c). Notably, spatially adjacent dsRNA segments tend to display nematic alignment, suggesting orientational ordering within these structured domains. To confirm this finding, we calculated the scalar order parameter S, defined as the largest eigenvalue of the local Q-tensor that describes the orientational order of anisotropic molecules in a nematic liquid crystal. Specifically, S was calculated from the orientation of each spatially adjacent pair of dsRNA segments and then averaged across all pairs. Across different salt concentrations, the average value of S ranged between 0.3 and 0.5 (Fig. 6i), which is characteristic of systems in the nematic phase.[72] As expected, the average S value decreased with increasing salt concentration, reflecting the enhanced flexibility of dsRNA under high-salt conditions. Contact probability map further revealed that dsRNA predominantly arises from short-range base-pairing events, although long-range interactions also contributed substantially (Fig. 6j). For instance, ~80% of dsRNA elements adopt hairpin structures with the sequence length smaller than 100 bp (Fig. 6k). Under physiological conditions, these clustered RNA hairpins with nematic alignment, which are interspersed by flexible ssRNA linkers, assemble into a tree-like architecture (Fig. 6l). This higher-order organization likely minimizes chain entanglement and enhances molecular recognition, while the ssRNA segments provide the flexibility needed for polymerase mobility and effective reverse transcription. Together, these results indicate that pgRNA adopts a partially ordered conformation comprising distinct dsRNA and ssRNA substructures, modulated by its interactions with the capsid inner surface.

To further characterize the structural organization of the pgRNA genome within the HBV capsid, we estimated the averaged maximum ladder distance (MLD) among the sampled structural ensemble of pgRNA in CG simulations, following previous works.[73,74] The average MLD value ranges between 110 and 120 bp under conditions where phase separation is prominent (Supplementary Fig. 22d). This value is significantly lower than the MLD of a randomly permuted single-stranded RNA of similar sequence length (~300 for a 3000-nt RNA), consistent with previous findings that viral RNAs are more compact than their randomized counterparts.[74,75] We also analyzed the interaction characteristics between RNA and CTD. Structural visualization showed that CTDs tend to be embedded between adjacent dsRNA segments, acting as bridges (Supplementary Fig. 20a, b). Although CTDs can also fit into the major or minor grooves of dsRNA, such specific interaction modes are not dominant. Notably, the CTD-RNA interactions are also highly dynamic, as evidenced by substantial variations in the number of CTD-RNA contacts within the clusters (Supplementary Fig. 20d). More detailed analysis revealed that the arginine-RNA ratio in the dense phase stabilizes at a characteristic value of ~1.05–1.11 across varying salt concentrations (Supplementary

Fig. 20c). This value is lower than the overall system average of 1.13, indicating a fractionation effect typical of multicomponent phase separation system.[56]

## Role of phase separation on long-range base-pairing and polymerase mobility

The reverse transcription of the pgRNA genome requires multiple template switching events within the HBV capsid, during which polymerase translocates between distant sites along pgRNA template. Prior experimental evidence has demonstrated that long-range base-pairing is essential for enabling the template switches.[76–79] In addition, efficient template switching requires sufficient mobility of the polymerase to navigate the pgRNA landscape and locate specific interaction sites. Thus, the LLPS-mediated structural organization of pgRNA genome within the capsid likely contributes to the molecular events that govern reverse transcription, particularly template switching. As an illustration, we performed additional simulations wherein the polymerase, an essential component for the reverse transcription process, was introduced to the pgRNA-encapsulated capsid (Fig. 7a, c). The resulting CTD-pgRNA condensates in the presence of polymerase mirrored those observed without it, forming a distinct hollow morphology characterized by coexisting low- and high-density regions (Fig. 7a and Supplementary Fig. 24a). Base-pair formation accompanied the CTD-pgRNA co-condensation (Supplementary Fig. 24b). In contrast, simulations in which CTD charges were neutralized abolished phase separation and eliminated the shell-like organization (Fig. 7b).

Under phase-separation conditions, contact probability maps revealed frequent intramolecular base pairing interactions (Fig. 7d). Notably, Abraham et al. demonstrated that interactions between the cis-acting element ε and the distal φ site facilitate template switching from ε to DR1,[69] highlighting the importance of long-range base pairing in reverse transcription. Here, we can observe an increase of long-range base pairing probability (Fig. 7d and Supplementary Fig. 23), which is essential for enabling the template switches as discussed above. In this study, the base pairs with sequence separation larger than 100 are classified as long-range base pairs. These long-range contacts were largely absent when phase separation was suppressed. Moreover, disruption of phase separation led to dense encapsulation of the polymerase by pgRNA, which significantly reduced polymerase mobility (Fig. 7e) and may impair its ability to efficiently locate template-switching sites. Collectively, these results demonstrate that phase separation not only facilitates structural organization of pgRNA genome but also enhances essential dynamics required for efficient template switching during HBV reverse transcription (Fig. 7f).

## Discussion

Phase separation has been observed during the life cycles of diverse viruses.[31–33,80–84] In several viruses, such as HIV-1, influenza A virus (IAV), and SARS-CoV-2, LLPS drives the co-assembly of viral RNA with capsid or nucleocapsid proteins at the early stage of their life cycle, facilitating nascent capsid formation.[34,85–88] Furthermore, phase separation can enhance the formation of viral inclusions during genome replication.[31] These virus-induced condensates can effectively exclude immune recognition factors, enabling viral replication without being detected by host defenses.[31] Here, we report phase separation occurring specifically within HBV capsid, a phenomenon not previously documented. We elucidate its potential biological significance in facilitating the structural organization of pgRNA genome and reverse transcription.

Compared to typical biomolecular phase separation observed in bulk solution or cellular environments, the phase separation of pgRNA genome within HBV capsid exhibits distinct features. Firstly, it occurs within a nanoscale confinement and the inner surface of the confinement space contributes key component (CTDs) for inducing phase separation. Notably, confinement modulation of phase behaviors is a subject of considerable interest in the field of physics studies.[89,90]

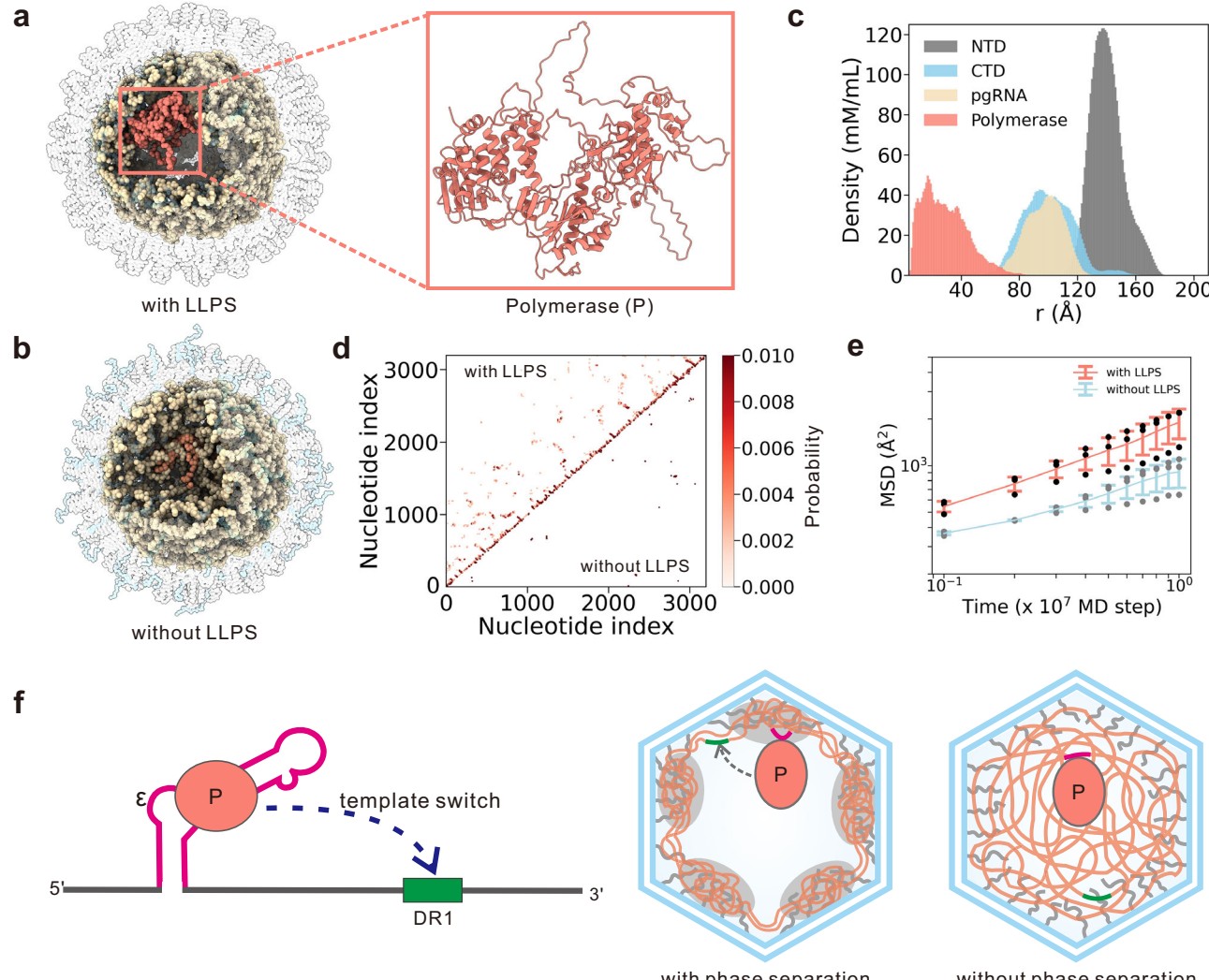

**Fig. 7 | Molecular simulations of pgRNA structural organization in the presence of polymerase.** Representative conformations of pgRNA-filled capsid in the presence of polymerase under conditions with (**a**) and without (**b**) phase separation. The cartoon representation of the three-dimensional structure of polymerase is also shown in (**a**). **c** Radial distribution functions of NTD (gray), CTD (blue), pgRNA (yellow), and polymerase (red) relative to the capsid center. **d** Contact map depicting base-pairing probability for the WT system (top) and CTD-Neu system (bottom). **e** Mean-squared displacement (MSD) of the polymerase as a function of time under the conditions with (red) and without (blue) phase separation. The Data are presented as mean +/− SD derived from 3 independent simulations. **f** Schematic illustrating possible role of LLPS in the reverse transcription. LLPS promotes the long-range base pairing and polymerase mobility (right panel), potentially contributing to the template switching events (i.e., polymerase switches its binding from the cis-acting regulatory element ε to another cis-element DR1; left panel). Source data are provided as a Source data file.

Molecules constrained within a confined environment may manifest new physical behaviors compared to those observed in bulk systems. For instance, wetting and layering can be observed on the confined space driven by the surface interactions. The findings in this work illustrated that the distinct physical features of phase separation in confinement space can be leveraged by HBV to facilitate biological functions. Particularly, the formation of the hollow spherical shell organization of pgRNA genome due to the surface induced CTD-pgRNA co-condensation can potentially contribute to the long-range pgRNA base pairing and enhanced mobility of polymerase, which is essential for the multiple template switches involved in reverse transcription. Secondly, the spherical shell organization renders a two-dimensional phase separation with coexistence of low- and high-density regions. These low density regions enable efficient transportation of ions and nucleotides across the capsid, which is critical for DNA synthesis and signal transduction during virus life cycle.[18] In addition, due to the icosahedrally symmetric arrangement of the core proteins, the ensemble-averaged density distribution of the CTD-

pgRNA condensates along the two-dimensional spherical shell exhibits icosahedral symmetry, which is consistent with cryo-EM observations. Intriguingly, the density distributions in the individual snapshots are highly heterogeneous without showing clear symmetry, suggesting symmetry breaking in the pgRNA genome organization of the single-particle HBV virions. Notably, two-dimensional phase separation has been typically observed in biological membrane, facilitating the assembly of membrane-associated compartments like integrin adhesion complexes[91,92]. Here, it is intriguing to observe that two-dimensional phase separation can also play important role in HBV virion replication, expanding the known biological contexts and functional roles of phase separation.

It is also worth emphasizing that the above discussed phase separation of the pgRNA genome within the HBV capsid differs from the macroscopic LLPS in free solution in the following two important aspects: i) the CTD is anchored to the inner surface of capsid and therefore cannot diffuse freely to form larger droplets through co-condensation with pgRNA; and ii) the finite size of the system makes

the condensate structure distinct from bulk-phase LLPS, where low- and high-density phases are typically well separated to form round-shaped droplets. At the same time, this CTD-RNA co-condensation shares key features with typical macroscopic LLPS, including the coexistence of high- and low-density regions, liquid-like condensate dynamics, and sensitivity to temperature and salt concentrations.

Under physiological conditions, the pgRNA genome organizes into higher-order, tree-like hairpin domains in which nematically aligned RNA hairpins are interspersed by flexible single-stranded linkers. These structured hairpin arrays may minimize chain entanglement and facilitate molecular recognition, while the ssRNA linkers confer the conformational flexibility required for polymerase translocation and efficient reverse transcription. Perturbations such as increased salt concentration tend to disrupt the pgRNA segregation, underscoring the environmental sensitivity of this balanced structural order and flexibility of pgRNA. The appropriate structural ordering of pgRNA genome is likely the result of evolutionary optimization to facilitate viral replication under physiological conditions. Remarkably, this LLPS-mediated pgRNA genome organization within capsid confinement mirrors fundamental features of eukaryotic chromatin folding[93,94], where LLPS drives the chromatin segregation into compartments of variable packing density with the cell nucleus confinement. In addition, the hierarchical organization of eukaryotic chromatin involves the local folding of genome into ordered topologically associating domains (TADs)[94]. Such higher-order chromatin architecture reconciles chromatin compactness with functional plasticity[95]. The resemblance between pgRNA organizations and chromatin architecture suggest that phase separation may constitute a universal strategy for structuring nucleic acid polymers across biology within confinement environment, despite the large differences in molecular mechanisms and length scales. Further mechanistic studies are needed to determine how this organization facilitates HBV reverse transcription.

The key role of phase separation in the structural organization of HBV pgRNA genome demonstrated in this work underscores the importance of physical principles in elucidating the viral life cycle. Indeed, substantial advances in understanding viral structure and assembly have been achieved through physics-based methodologies, which have successfully explained a wide range of experimental observations[96]. For instance, by using a mean-field theory, Zandi and coworkers have illustrated the effects of charge distribution of capsid protein and RNA topology on the amount of packaged RNA based on the genome configurational entropy and electrostatics[97,98]. Base-pairing in RNA can trigger a conformational transition of RNA genome from an extended coil to a compact globule[99]. In addition, kinetic and thermodynamic analysis of viral capsid assembly by Hagan et al revealed the importance of genome tertiary structure in capsid assembly and elucidated the key factors contributing to overcharging effect[73]. Collectively, these studies highlight the close interplay between RNA secondary structure, the electrostatic regulation, and the optimal length of the encapsulated genome, emphasizing the value of physics-based approaches in elucidating virion organization.

In summary, by integrating multiscale MD simulations with biochemical assays, we demonstrate that HBV pgRNA harnesses liquid-liquid phase separation within the capsid to sculpt its own architecture and facilitate reverse transcription. Electrostatic interactions between the disordered CTDs and pgRNA are the primary driving force of phase separation, which leads to a hollow spherical shell organization of CTD-pgRNA condensates with coexistence of low- and high-density regions. LLPS drives the establishment of microphases composed of nematically aligned RNA hairpin arrays interspersed by clusters of flexible ssRNA linkers, achieving an optimal balance of structural order and dynamic flexibility essential for function. Although ensemble-averaged density profiles recapitulate the icosahedral symmetry of HBV capsid, individual simulation snapshots reveal pronounced

heterogeneity, signifying symmetry breaking at the single-virion level. The phase separation promotes the establishment of long-range base-pairing interactions and enhance the mobility of the polymerase, which are pivotal in reverse transcription. Together, these findings uncover a capsid-confined phase separation mechanism that orchestrates viral genome organization and dynamics, paving a new avenue for antiviral intervention.

## Methods
### All atom MD simulations
The HBV capsid was built from an atomic model resolved by cryo-EM (PDB code: 3J2V), where the missing residues in CTD was reconstructed by Modeller[100]. The initial structure of the 3200-nt pgRNA was generated using the Nucleic Acids Builder (NAB) script in Ambertools[101] with the wild-type pgRNA sequence. The pgRNA sequence without the polyA tail was retrieved from GenBank (accession number V00866.1). The 5′- and 3′- termini of the pgRNA sequence were identified by referring to the information documented in a previous study[102]. The pgRNA conformation generated by NAB without pre-assigned secondary structures was subsequently relaxed though MD simulations at elevated temperature to produce a more randomized initial structure, thereby minimizing potential bias from the starting configuration. As a control, we also used an initial structure of pgRNA with rich secondary structures modeled by AlphaFold3[103]. Finally, the pgRNA was further relaxed by applying a spherical wall potential in MD simulation, resulting in a relatively compact structure with dimensions suitable for integration into the capsid.

The pgRNA-containing capsid was solvated within a periodic box (390 Å × 390 Å × 390 Å) of 1,812,893 TIP3P water molecules. $Na^+$ and $Cl^-$ were added in the box to neutralize the charge and achieve a salt concentration of 150 mM, leading to a system containing 6,278,802 atoms. Here the salt concentration corresponds to the average salt concentration of the whole system. The amber14sb_$OL$21 force field was used, which combines the amber14sb force field for protein and $\chi_{OL3}$ force field for RNA[104,105]. To assess the force field dependence of the results, we performed additional all-atom MD simulations employing the amber14sb_$OL$15 force field with NBFIX correction[35,36]. The electrostatic interaction was treated by the particle mesh Ewald (PME) method with a distance cutoff of 10 Å. The same cutoff value was also used for the van der Waals (VDW) interactions. The MD time step was 2fs and all bonds involving hydrogen atoms were constrained using the LINCS algorithm. The initial structure was first minimized for 100,000 steps using the steepest decent algorithm, which was followed by a 5-ns equilibrium simulation with an NPT ensemble at 1 atm and 300K. The production simulations with the length of 100 ns were then performed at 1 atm and 300K, with atomic coordinates recorded every 10 ps. V-rescale thermostat and Parrinello-Rahman barostat were used for temperature and pressure coupling,[106,107] with the time constants of 0.1 ps and 2 ps, respectively. The simulations were conducted with Gromacs (version 2020.4).[108,109]

Two additional all-atom molecular dynamics simulations were performed to characterize phase separation on a two-dimensional planar membrane composed of CTD segments. One system contained a 15 × 16 array of CTD chains (240 chains total), while the other consisted of a 10 × 10 array (100 chains). The initial all-atom configuration was generated by structure reconstruction from CG structure in which $A_{15}$ RNA molecules were uniformly distributed across the anchored CTD layer.[110,111] The corresponding systems contained 219 and 71 RNA chains, respectively. Each setup was solvated in a periodic box with TIP3P water molecules, and $Na^+$ and $Cl^-$ were added to neutralize the system and achieve corresponding salt concentrations. The amber14sb_$OL$15 force field with NBFIX correction was used, with all other settings consistent with the above pgRNA-containing capsid simulations. During production simulations, the temperature was first raised to 500 K and then gradually reduced to 300 K over 20 ns.

Subsequently, production simulations were conducted for the two systems for 100 ns and 200 ns, respectively, in the NPT ensemble at 1 atm and 300 K, with atomic coordinates saved every 10 ps.

To identify Watson-Crick base pairs, we applied geometric criteria to complementary nucleotides with a sequence separation greater than one. A pair of nucleotides was considered to form a base pair when the bases of two complementary nucleotides established hydrogen bonds, defined by a donor-acceptor heavy atom distance of less than 3.5 Å and the N-H-O or N-H-N angle exceeded 120°.

### Residue-resolved coarse-grained MD simulations

The residue-resolved coarse-grained models were employed to characterize the protein and nucleotide acids involved in the HBV virion. For the capsid protein, each residue was represented by one particle, which is centered at the Cα position. The atomic interaction-based coarse-grained model with flexible local interactions (AICG2+)[47,48] was used to describe the dynamics of the folded NTD domain of capsid protein, while the hydrophobicity scale (HPS) model[57] was employed to characterize the interactions involving the intrinsic disordered CTD domain. The AICG2+ model has the following function form:

$$V(\mathbf{R}|\mathbf{R_0}) = V_{bond} + V_{flp}^{loc} + \lambda V_{SB} + V_{exv}. \tag{1}$$

The first term, the bond potential ($V_{bond} = \sum_i k_b(\mathbf{r_{i,i+1}} - \mathbf{r_{i,i+1}^0})^2$) describes covalent connectivity between consecutive residues. The second term, the flexible local potential ($V_{loc}^{flp} = V_a^{stat} + V_d^{stat}$), characterizes the sequence dependent flexibility and secondary structure propensity of the protein chain. Here, $V_a^{stat}$ and $V_d^{stat}$ denote statistical potentials for bond angles and dihedral angles, respectively, derived from a loop-segment dataset[112]. The third term $V_{SB}$ represents the structure-based potential,[113] defined as:

$$V_{SB} = \sum_{j=i+2} \epsilon_{1,3}^{ij} exp(-\frac{(\mathbf{r_{ij}} - \mathbf{r_{ij}^0})^2}{2W_{1,3}^2}) + \sum_{j=i+3} \epsilon_{1,4}^{ij} exp(-\frac{(\phi_i - \phi_i^0)^2}{2W_\phi^2})$$
$$+ \sum_{j>i+3}^{native} \epsilon_{nloc}^{i,j}[5(\frac{\mathbf{r_{ij}^0}}{\mathbf{r_{ij}}})^{12} - 6(\frac{\mathbf{r_{ij}^0}}{\mathbf{r_{ij}}})^{10}] \tag{2}$$

This potential combines local interactions with non-local interactions. $\mathbf{r_{i,j}}$ denotes the distance between residues $i$ and $j$ in a given conformation, while $\mathbf{r_{i,j}^0}$ represents the corresponding distance in the reference structure. Similarly, $\phi_i$ corresponds to the dihedral angle defined by four consecutive residues ($i$, $i+1$, $i+2$, and $i+3$), with $\phi_i^0$ being its reference value. The force field parameters $k_b$, $\epsilon_{1,3}^{ij}$, $\epsilon_{1,4}^{ij}$, $\epsilon_{nloc}^{ij}$, $W_{1,3}$, and $W_\phi$ were assigned their default values from the GENESIS 1.7.1 software package.[114–116] In this study, the cryo-EM resolved structure (PDB code: 3J2V)[117] was used as the reference of the capsid structure. The final term, $V_{exv}$, accounts for excluded volume interactions between all non-native residue pairs. To better capture the flexibility of the HBV capsid, we introduced a scaling factor $\lambda$ for the structure-based potential. At $\lambda = 0.7$, the coarse-grained model accurately reproduced the average root mean square fluctuation (RMSF) of HBV capsid protein dimers obtained from previous atomistic molecular dynamics (MD) simulations.[18]

For the nucleotide acids, the three-site-per-nucleotide model (3SPN.2)[49] was adopted. Each nucleotide was simplified as three particles representing the phosphate group, the sugar group and the base group, respectively. To accurately depict the flexibility of the RNA molecule, the magnitude of the dihedral energy term was adjusted according to the experimentally measured persistence length value for a polyU sequence ($U_{40}$) (Supplementary Fig. 25)[118]. The Debye–Hükel type electrostatic potential was used to characterize the salt-concentration-dependent interactions of charged particles. It is worth mentioning that with such an implicit solvent model, the

Donnan equilibrium cannot be explicitly captured. While neglecting this effect in the coarse-grained model may limit the quantitative interpretation, the major results are expected to be less sensitive to this limitation as the positive charges of CTDs and negative charges of pgRNA are roughly balanced.

In the simulations of polymerase-containing system, the full-length structure of polymerase was modeled from AlphaFold2[119]. Langevin dynamics was used to control the temperature. The time step was set to 0.01 ps. The friction parameter of the Langevin barostat was selected as 0.1 ps⁻¹. The length of the simulation was set to $1 \times 10^8$ MD steps and three independent simulations were performed for each case with different random seeds. All simulations were conducted with GENESIS (version 1.7.1)[114–116].

In the coexistence simulation of CTD peptides and $A_{15}$ RNA molecules, a slab simulation methodology was implemented. The system was placed in a periodic box with a long z-axis (194 Å × 194 Å × 3600 Å), which contained 216 CTD chains and 216 $A_{15}$ chains. The initial compact conformation was derived from a shrinking simulation wherein the box was compressed to dimensions of 194 Å × 194 Å × 194 Å over $1 \times 10^6$ MD steps, at the temperature of 300 K and salt concentration of 150 mM. The simulation lasted for $1.0 \times 10^8$ MD steps. All other simulation parameters matched those described above.

Here, a contact was defined between two CG particles separated by ≤6.5 Å. Using this definition, contacting nucleotides were grouped into clusters. Nucleotides were then classified as "free" or "condensed" based on cluster size. In addition, a Watson-Crick base-pair was assigned when the distance between the base beads of two complementary nucleotides (with a sequence separation greater than one) was less than a cutoff value of 6.5 Å.

To estimate phase diagrams, the simulation space was divided into discrete bins, and the density within each bin was calculated. Dense (high-density) and dilute (low-density) phases were defined based on the average density, using threshold scaling factors of 1.2 and 0.8. Specifically, bins with densities exceeding 1.2 times the average were assigned to the dense phase, while those below 0.8 times the average were assigned to the dilute phase.

### Experimental materials

CTD-peptide and RNA polyA ($A_{15}$) were synthesized by GenScript, Nanjing, China. 1,6-hexanediol (1,6-HD) was purchased from Aladin, Shanghai, China. The peptide was dissolved in TE buffer (10 mM Tris-HCl, pH 7.5 and 2 mM EDTA). The concentration of CTD-peptide was double checked from the absorbance at 280 nm using a Nano-500 Micro-spectrophotometer (Allsheng, Shanghai, China). The $A_{15}$ was also dissolved in TE buffer and the concentration was measured with the same spectrophotometer. All solutions were filtered using 0.22 μm syringe filters before use.

The microscope coverslips for the observation of droplets were obtained from VWR. Before observation, the coverslips were ultra-sonic cleaned with 5% detergent for 30 minutes, followed by 2 cycles of mPEG-SVA (Laysan Bio., USA) passivation treatments to reduced nonspecific bindings. The coverslips were dried in a drying oven afterwards. The observation wells (radius ~3 mm, height ~5 mm) were home-made using plastic rings glued to the passivated surface.

### Bright field microscopy imaging of droplets

A stock solution of CTD-peptide (~1 mM) and $A_{15}$ (~200 μM) was used to prepare the sample. Total volume of a 15 μl solution containing 10 μM $A_{15}$ and variable concentrations of CTD-peptide in TE buffer (pH = 7.5) with different salt concentrations was mixed in a centrifuge tube, and transferred to the observation well immediately after a brief vortex-mix. The samples were observed after incubation for 30 min. Images were obtained using an inverted microscope equipped with a Basler

camera (NexcopeNIB610-FL) and official Basler software. All experiments were repeated 3 times.

## Atomic force microscope experiments

No. 1.5 glass coverslips (VWR) were used as substrates. The surfaces were first functionalized with (3-aminopropyl) triethoxysilane (APTES, Fisher Scientific) to generate an amine-terminated layer. Subsequently, the heterobifunctional crosslinker Maleimide-PEG2000-SVA (20 mg dissolved in 60 μL of buffer) was conjugated to the surface amines via its NHS ester (SVA) group. A cysteine-terminated peptide (CTD, 5 mM) was then coupled to the maleimide end of the tethered PEG chains. This conjugation step was performed in the presence of 2 mM TCEP as a reducing agent. Finally, 10 μL of a 5 μM RNA sample ($A_{15}$) was deposited onto the peptide-modified surface, incubated for 5 min, rinsed thoroughly with 2 mL of deionized water, and dried under a gentle stream of nitrogen gas. Atomic force microscopy (AFM) was performed using a Bruker Dimension Icon system. Imaging was conducted in tapping mode in air using NSG30 probes (Golden Silicon Probes) with a nominal spring constant of 20–100 N/m. The images were captured at a scan size of $1 \times 1 \mu m$ with a scan rate of 1 Hz. All experiments were performed at room temperature.

## Reporting summary

Further information on research design is available in the Nature Portfolio Reporting Summary linked to this article.

## Data availability

All source data generated in this study have been deposited in the Figshare database (https://doi.org/10.6084/m9.figshare.29605127). The input files of MD simulation and representative trajectory files are available at https://box.nju.edu.cn/d/eb918ffb092246cf8971/. Source data are provided with this paper.

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

## Acknowledgements

The authors gratefully acknowledge Dr. Xin Wang of BASF for his technical support with the Atomic Force Microscopy (AFM) experiments. This work was supported by National Natural Science Foundation of China (Grant Nos. 12574224 to W.L., 12090052 to W.W., and 12347102 to W.L.), Basic Research Program of Jiangsu Province (BK20253050 to W.L.), and the grant of Wenzhou Institute, University of Chinese Academy of Sciences (WIUCASQD2021010 to W.L., WIUCASQD2022036 to Y.B.). The authors also thank the support from HPC Center of Nanjing University,

e-Science center of Nanjing University, Nanjing Kunpeng&Ascend Center of Cultivation, and the Nanjing Key Laboratory for Cardiovascular Information and Health Engineering Medicine (funded by the Nanjing Municipal Health Commission) and its Jiangsu counterpart.

## Author contributions

W.L. and W.W. conceived the ideas and designed the work. Y.B. and Y.H. conducted the molecular dynamics simulations. H.P. and J.M. carried out the experiments. Y.W. and Y.C. contributed to the discussion. Y.B., W.L. and W.W. co-wrote the manuscript.

## Competing interests

The authors declare no competing interests.

## Additional information

**Peer review information** : *Nature Communications* thanks Aleksei Aksimentiev who co-reviewed with Kush Coshic, and the other, anonymous, reviewer(s) for their contribution to the peer review of this work. A peer review file is available.

