## [Transparent Peer Review file · Nature Communications]

Structural organization of HBV pgRNA genome driven by phase separation in capsid confinement

Corresponding Author: Professor Wenfei Li

Version 0:

Reviewer comments:

Reviewer #1

(Remarks to the Author)

The authors present a comprehensive and well-structured study on the structural organization of the hepatitis B virus (HBV) pregenomic RNA (pgRNA) genome, driven by phase separation within the confined capsid environment. Employing multi-scale molecular dynamics simulations alongside biochemical assays, they demonstrate that the pgRNA–C-terminal domain (CTD) condensate exhibits coexisting low- and high-density regions, leading to the formation of hollow, shell-like condensates along the arginine-rich inner surface of the capsid. These interactions are predominantly electrostatic, consistent with expectations for RNA viruses. While the study is thorough, many aspects of the manuscript remain speculative, necessitating further justification of the underlying hypotheses and in particular the modeling approaches used. Provided that the authors adequately address these concerns, this work will be suitable for publication in Nature Communications, as it offers valuable insights of broad relevance.

Major concerns:

- The description of the molecular dynamics methods lacks critical details necessary for clarity and reproducibility. For instance, the modeling of the pgRNA genome is insufficiently described; the only mention is at line 298, stating that “the initial structure of a 3200-nt pgRNA was created by AmberTools.” The authors should specify which AmberTools feature or protocol was used to generate the random placement of RNA beads. Moreover, before relying on a randomly placed pgRNA as a foundational assumption, the authors must justify this choice by referencing published studies that support and validate this approach.
- The authors have used the Berendsen algorithm for both temperature and pressure coupling. This is dangerous because it is well known that the Berendsen coupling does not provide adequate Boltzmann sampling. For this reason, GROMACS even warns against using Berendsen coupling for anything beyond initial equilibration. Redoing simulations would be beyond the scope, and hence, the authors must provide a reasonable reason why their findings would be unaffected.
- Looking at the all-atom structure of the RNA (input files provided at the link) I found the RNA to be completely unstructured. This is an important assumption that needs to be discussed: that the authors (i) model the RNA as a random polymer and (ii) the RNA is unstructured, i.e. devoid of any secondary structure to begin with. Both assumptions require a reference. Looking at the trajectory, it was also very difficult to convince myself that Watson-Crick base-pairing was spontaneously observed in the pgRNA genome, something I was curious having seen the unstructured RNA at the beginning of the simulation. Could the authors pin-point to specific residues that validate the observation of RNA base-pairing? This is an important result that has a section on its own (Figure 6). How exactly are base-pairs defined would be an important information to mention in the methods section. This needs to be done for both all-atom and coarse-grained (CG) trajectory analysis (see line 192). The definition of base pairs in the CG model may overestimate true Watson-Crick base pairing.
- Line 70: The “pgRNA quickly condensed along the inner surface of the capsid”: Could this be because of bad initial configuration of the RNA whereby any internal stabilization via base-pairing is completely dominated by strong nucleic acid-Arginine interactions? A movie of the initial spherical confinement to this shell-like arrangement would be a useful addition.
- Traditionally, MD force fields can overestimate interactions between highly charged moieties, that includes Arginine residues binding strongly to the phosphate groups on nucleic acids. Conventional practice prevails the use of specific

NBFIx or charge rescaling to tackle this, none that were used in this study. Could the authors comment on the choice of force field, whether it can adequately represent these interactions. This issue is particularly critical because the pgRNA-CTD condensate formation is driven by electrostatics, and overestimating these interactions could result in an artificial state that mimics condensate properties. Nonetheless, the slab simulations provide a robust representation.

- Line 114: Using only a single 50nt segment out of the 3200nts is questionable as “randomly selected”. Also Fig. S2b’s caption mentions this segment as nucleotides 1-50. If so, the “randomly selected” in line 115 should be removed. The RNA is confined and one may argue that the true MSDs can be very sensitive to the choice of the fragment, in particular due to the observation of both voids and high density regions. The same analysis should therefore be based on an average over several such fragments. Why can’t the analysis be done for all such possible fragments?
- Line 174-177: The authors mention that the RNA density is much lower at the threefold symmetry sites, compared to fivefold and sixfold. This is an interesting result. Could the authors comment on why this may be so? Also, it is not clear whether the analysis done in Figure 5 used only the all-atom trajectories or does it also include coarse-grained trajectories. Accordingly the caption needs to be updated, that only says “MD simulations.”
- Line 193-195: The authors argue that a marked increase in the number of base pairs indicates that LLPS enhances the pgRNA secondary structure. Could the increase of base pairs be just due to a bad initial structure of the pgRNA that is devoid of secondary structure at the beginning? If so, that LLPS enhances this formation would be purely speculative.
- How is long-range base pairing defined? This is hard to convince from nucleotide contacts plots (Fig. 6j) because nucleotides distant w.r.t. their index/residue may still be spatially located in relative proximity and vice versa. Accordingly, the text may need to be updated (for e.g. line 215). The use of the word “significant” on line 235 is speculative.

Minor concerns:

- Line 17: I would recommend citing Bloomfield, V. A. DNA condensation by multivalent cations. *Biopolymers* 44, 269–282 (1997), as the first to suggest “significant electrostatic repulsions”.
- Figure 2c-f: The colors are not defined in the caption.
- Figure 4a: It is not clear how the anchored CTD is modeled. Is it all the 37 residues of the CTD? From the image, the anchored NTD is not fully modeled and it is unclear where the anchor point is. This should be updated in line 152 (“N-termini fixed”).
- Figure 5a: The isosurface density scale is missing. From the image it may falsely appear that there are no RNA residues in the threefold symmetry sites, whereas this is likely only due to the isosurface density used to create the image.
- Figure 5c: Color scale is missing
- Line 199-203: Are these simulations all-atom or coarse grained?
- The MSD plots (Figure 2k and 7e) suggest the diffusion is very much Brownian like (straight lines at longer lag times). This is a bit counter-intuitive at longer lag times because of the capsid confinement.
- Line 92: The “an” should be “a”

Reviewer #2

(Remarks to the Author)

The manuscript describes results of a detailed study, mostly by means of multiscale molecular dynamics computer simulations and partly by experimentation, of the structural organization and condensation of the pregenomic RNA in the capsid of the virus HBV. The authors focus on the role of the positively charged, structurally disordered C-terminal RNA binding domain of the coat protein, often called Arginine Rich Motif or ARM, in relation with their spatial confinement in the capsid. Parenthetically, in the manuscript the ARMs are called CTD, short for C terminal domain.

The main point the authors try to convince the readers of is that the structural arrangement of the RNA in HBV is driven by phase separation, arguing that viruses exploit what is known as liquid-liquid phase separation or LLPS. LLPS has attracted a lot of attention in biophysical literature recently. Often (but not always) driven by interaction of, e.g., RNAs with positively charged, structurally disordered proteins, and I understand why seeing a correspondence between virus assembly and LLPS in biology is such an attractive one.

I find the paper hugely interesting, and I am particularly impressed by the comprehensiveness of the simulation study. The findings are highly significant, and, if accurate, would further our understanding of the structure of viruses and appeal to a broad readership. Also, the paper is very well written, well-structured and easy to read. I do find pushing the LLPS narrative so forcefully as is done in the manuscript at times a little too much. As a matter of fact, I am not yet convinced that the case made by the authors actually hold water, irrespective of the interesting findings they present.

Some might view this a semantic issue, but in my view absorption of what in the end is a polymer (the ssRNA) into a structured polymer brush (the ARMs emanating into the cavity of a capsid) is not the same as macroscopic phase separation, in the same way that condensation of a gas in bulk is not the same as condensation of a gas onto an adsorbing surface. Strictly speaking, the assembly of coat proteins around an RNA cannot represent a true phase transition on account of its finite size, noting that the droplet phases typically found in LLPS must be transient. The physics of adsorption of RNA onto immobilized ARMs part of a capsid is very different from the co-condensation of ARMs with RNA in free solution – the authors are aware of this but seem to downplay this.

There are a few other issues that I have with the manuscript that need to be addressed before I would advise in favour of publication.

1. There is a lot of theoretical and simulation literature on the structure of (model) RNAs in virus capsids, and the readers should be aware of that. I am referring to, e.g., the work of Hagan, Zandi and Podgornik to name but a few. Of particular importance for the topic of the manuscript is the work of Hagan in *eLife* 2013; 2: e00632, and Zandi and Podgornik in *Phys. Rev. E* 2014; 89: 032707 and Zandi in *Phys. Rev. E* 2020; 102: 062423. These, and other works, highlight the importance of the localization of the positive charges, but also the configurational (base-pairing) response of the RNA to the binding to the ARMs. See also the review paper of Zandi and collaborators, *Physics Reports* 2020; 847: 1–102.

2. I do not believe that the observed coexistence of high- and low-density regions in the absorption layer near the ARMs indicates phase separation, as is implied on page 3, lines 77 and 78. The difference in RNA density in the cavity and that in the absorption layer would be in closer to correspondence to complex coacervation of ARMs and RNA, discussed also in the main text. I am very skeptical about the findings from the simulations that phase separation takes place in the absorption layer in the virus (fig. 2), and on the flat surface with end-grafted ARMs (fig. 4) unless the salt concentration is sufficiently high. The reason is that this leaves the regions with low RNA density with nearly naked ARMs, so without the benefit of attractive interaction with the RNA whilst the higher-density regions must suffer from higher self-repulsion of the RNA. No experimental verification has been done, as was with the bulk phase separation of RNAs and ARMs (fig. 3). I would strongly recommend experimental verification, knowing full well that this is a major endeavor. The discovery of dewetting of a polyanion such as RNA adsorbed on a polycationic surface would represent a major surprise in the field.

3. The in-layer phase separation shown in fig. 4 was studied for different ionic strengths, but only a single RNA length. It would be useful to investigate how the length of the RNA influences this phase separation. Also, the phase separation, if real, should lead to inhomogeneous distribution of mobile ions. Not much is said about this in the manuscript. Note also, that if phase separation indeed takes place, the high- and low-density regions should increase in size in time and coarsen, e.g., by coalescence. Does coarsening take place?

4. The structure of the RNA in particular in relation to base pairing is discussed on page 9 and 10 (figure 6). There is quite some work on the structure of viral ssRNAs, and why they are different from random RNAs or cellular RNAs. See, e.g., the work of Gelbart and others on this topic in *PLoS ONE* 2014; 9: e105875 and *Biophys. J.* 2017; 113: 339-347, to cite a few of many papers these authors wrote on this topic. Also, theoretical work (e.g. of Zandi, see earlier) indicates that interaction with the ARMs changes the base pairing. It would be interesting to get more information on the structure of the RNA in the virus, such as maximum ladder distances, length distributions of hair pin structures, and so on. The authors also mention nematic alignment of clustered hairpins but do not elaborate on how this was established: by eye or by calculating the largest eigenvalue and associated eigenvector of the local Q tensor? This would also produce a director field configuration – in that case one would expect four half-integer disclination defects to appear. There is quite a bit of literature on this.

5. Viruses tend to be overcharged, that is, there is more negative charge on the RNA than needed to compensate for the positive charges on the ARMs in the virus. This causes a Donnan equilibrium to establish itself for the mobile ions inside and outside the virus, and between the empty core of the lumen of the capsid and the region where ARMs and RNA interact. This equilibrium takes time to set up, and I wonder if the all-atom simulations are sufficiently long to let this equilibrium establish itself. Also, the Debye-Hückel potential used by the authors for the coarse-grained simulations most certainly cannot deal with this kind of phenomenology. It would be good to see if for physiological conditions Donnan effects can indeed be neglected by calculating time dependent ion density distributions in the atomistic simulations.

Reviewer #3

(Remarks to the Author)

Bian et al computationally show that interaction of RNA with the basic C terminal domain of the HBV core protein results in a condensed phase. The condensate resembles the hollow shell of density seen in image reconstructions of HBV with the pregenome and with random RNA from the expression system. In the condensed phase the RNA is able to diffuse. As one would expect, neutralisation of phosphates allows base pairing. Unsurprisingly, the dsRNA segments shows nematic organisation. Unsurprisingly, the heterogeneity of the layer follows the symmetry of the capsid. To support their *in silico* studies, the authors demonstrated that isolated CTDs were able to form a phase when mixed with 15 nucleotide poly A oligomers.

There are several issues the authors need to address. The approach to adding pgRNA to a capsid is novel; do all trajectories lead to the same result? How much do they differ? While the peptide – RNA oligomer complex is experimentally shown to form a liquid-liquid phase separation, the two dimensional nature of the inner layer of the capsid stretches the definition of a phase separation. This needs to be addressed. The phase needs to be better described. Do the high RNA density clusters show a characteristic arginine-RNA ratio? How fluid are the CTD-RNA clusters? How fluid are the dsRNA

segments? If sufficiently stable, they will halt reverse transcription. A quantitative view of their stability will be appreciated. One of the advantages of molecular dynamics, even with coarse graining, is the ability to visualize molecular contacts. How do the CTDs interact with the RNA? Do they interdigitate between strands? Do they fit in major grooves of dsRNA? The authors appear to use the HBV pgRNA in their study but do not identify the sequence used. It is not clear if they have a polyA tail. The means of adjusting ionic strength are not clearly defined. Are the authors holding the capsid interior to the stated bulk ionic strength? Is the exterior held at the stated ionic strength? If the latter, what is the interior ionic strength or total number of ions?

There are several papers that are germane to this study that ought to be discussed to provide deeper context. Most importantly, Harati Taji et al (DOI: 10.1021/jacs.1c12439) showed that HBV-RNA interactions were highly fluid in NMR studies. How does your study conform to Harati Taraji predictions? Chu et al (DOI: 10.1128/JVI.03235-13) showed that HBV CTDs could act as RNA chaperones. Is this consistent with your results? Do dsRNA duplexes readily break in your simulations? Patel et al (DOI: 10.1038/nmicrobiol.2017.98) identify putative packing sites; are they folded in your study and how would a stable fold affect your predictions. Abraham and Lob (DOI: 10.1128/JVI.80.9.4380-4387.2006) predict long-range base pairing; how would such base pairing bias the packaging of RNA.

Version 1:

Reviewer comments:

Reviewer #1

(Remarks to the Author)

The authors have done an incredible job with addressing all comments from the previous round of review. I recommend publication in its present form.

Reviewer #2

(Remarks to the Author)

I am satisfied with the extensive responses of the authors to the queries and comments of all three reviewers, and with the substantial revision of the manuscript that includes a discussion of additional studies that makes it an outstanding one.

Reviewer #3

(Remarks to the Author)

This revised manuscript addresses this reviewer's questions. With this revised ms, I better understand the effect of phase separation on polymerase location, the molecular interaction between CTDs and RNA, and the effect of CTDs on RNA mobility. The agreement between computation and experimental observation is convincing, notably the asymmetric reconstruction in reference 66, and will lead to a far better understanding of the process of in-capsid reverse transcription peculiar to HBV. This mechanism of RNA packaging is also likely relevant to many other small RNA viruses.

There are several minor typos and clarifications to address:

15 "rich in" instead of "rich of"

40 some HBV serotypes have a 183 residue length

75 notably

88 delete "become"

91 are the Na⁺ concentrations in units of M or mM?

221 "AFM morphology" or "atomic force microscopy (AFM)"

235, 237, etc there are no sixfold axes. There are quasi-sixfold vertices

298 define Q-tensor

304 provide units for "100". Is it Å, bp?

312 provide units for ladder length (Å or bp)

337 units

Reviewer #4

(Remarks to the Author)

Responses to Reviewer's Questions and list of changes

Reviewer #1 (Remarks to the Author):

The authors present a comprehensive and well-structured study on the structural organization of the hepatitis B virus (HBV) pregenomic RNA (pgRNA) genome, driven by phase separation within the confined capsid environment. Employing multi-scale molecular dynamics simulations alongside biochemical assays, they demonstrate that the pgRNA–C-terminal domain (CTD) condensate exhibits coexisting low- and high-density regions, leading to the formation of hollow, shell-like condensates along the arginine-rich inner surface of the capsid. These interactions are predominantly electrostatic, consistent with expectations for RNA viruses. While the study is thorough, many aspects of the manuscript remain speculative, necessitating further justification of the underlying hypotheses and in particular the modeling approaches used. Provided that the authors adequately address these concerns, this work will be suitable for publication in Nature Communications, as it offers valuable insights of broad relevance.

Reply:

We thank the reviewer for the insightful comments.

Major concerns:

1) The description of the molecular dynamics methods lacks critical details necessary for clarity and reproducibility. For instance, the modeling of the pgRNA genome is insufficiently described; the only mention is at line 298, stating that “the initial structure of a 3200-nt pgRNA was created by AmberTools.” The authors should specify which AmberTools feature or protocol was used to generate the random placement of RNA beads. Moreover, before relying on a randomly placed pgRNA as a foundational assumption, the authors must justify this choice by referencing published studies that support and validate this approach.

Reply:

We thank the reviewer for the thoughtful comments and suggestions. In this revised manuscript, we have expanded the Methods section to provide additional details on the MD simulation steps. In this work, the initial structure of pgRNA was generated using the Nucleic Acids Builder (NAB) script in AmberTools. The pgRNA sequence corresponds to the wild-type sequence obtained from GenBank (accession number V00866.1), with the 5'- and 3'-terminal regions set according to the work by Haines et al. (*J. Mol. Biol.* 2007, 370:471–480). The NAB-generated pgRNA conformation was subsequently relaxed through molecular dynamics (MD) simulations at elevated temperatures to yield a more randomized initial structure, thereby reducing potential bias from the starting configuration. We have added the above description of the molecular simulation methods to further enhance clarity and reproducibility in this

revised manuscript.

Corresponding changes:

1. (page 3, line 71) The following sentence was refined to clarify the initial structure of MD simulations in this revised manuscript:

“In the initial configuration of the MD simulations, the pgRNA with the wide-type sequence was placed inside the capsid in a compact conformation without pre-assigned secondary structures (Fig. 1c and Methods).”

2. (page 16, line 418) The following sentences were added in the Methods section to further clarify the used tool and pgRNA sequence in the construction of the initial structure of pgRNA model:

“The initial structure of the 3200-nt pgRNA was generated using the Nucleic Acids Builder (NAB) script in Ambertools¹⁰¹ with the wild-type pgRNA sequence. The pgRNA sequence without the polyA tail was retrieved from GenBank (accession number V00866.1). The 5'- and 3'- termini of the pgRNA sequence were identified by referring to the information documented in a previous study¹⁰². The pgRNA conformation generated by NAB without pre-assigned secondary structures was subsequently relaxed through MD simulations at elevated temperature to produce a more randomized initial structure, thereby minimizing potential bias from the starting configuration. As a control, we also used an initial structure of pgRNA with rich secondary structures modelled by AlphaFold3¹⁰³. Finally, the pgRNA was further relaxed by applying a spherical wall potential in MD simulation, resulting in a relatively compact structure with dimensions suitable for integration into the capsid.”

3. (page 16, line 427) To enhance clarity and reproducibility, more details of MD simulations were added in the Methods section:

“The pgRNA-containing capsid was solvated within a periodic box (390 Å × 390 Å × 390 Å) of 1,812,893 TIP3P water molecules. Here the salt concentration corresponds to the average salt concentration of the whole system. To assess the force field dependence of the results, we performed additional all-atom MD simulations employing the amber14sb_OL15 force field with NBFIX correction.^{35, 36} The initial structure was first minimized for 100000 steps using the steepest decent algorithm, which was followed by a 5-ns equilibrium simulation with an NPT ensemble at 1 atm and 300K. The production simulations with the length of 100 ns were then performed at 1 atm and 300K, with atomic coordinates recorded every 10 ps. V-rescale thermostat and Parrinello-Rahman barostat were used for temperature and pressure coupling,^{106, 107} with the time constants of 0.1 ps and 2 ps, respectively.”

2) The authors have used the Berendsen algorithm for both temperature and pressure

coupling. This is dangerous because it is well known that the Berendsen coupling does not provide adequate Boltzmann sampling. For this reason, GROMACS even warns against using Berendsen coupling for anything beyond initial equilibration. Redoing simulations would be beyond the scope, and hence, the authors must provide a reasonable reason why their findings would be unaffected.

Reply and corresponding changes:

We thank the reviewer for the kind comment. Upon re-examining the input file of the all-atom MD simulations, we realized that the description of the temperature and pressure coupling in the original version of the manuscript was not entirely accurate. For temperature coupling, we employed the v-rescale thermostat, which is an improved version of the Berendsen thermostat by introducing a stochastic term to the rescaling factor. It can better generate correct sampling of the canonical ensemble (*J. Chem. Phys.* 126, 014101, 2007). For the pressure coupling, we used the Parrinello-Rahman algorithm, which is a widely adopted option of pressure coupling in GROMACS. We apologize for any confusion caused by the earlier description. In the coarse-grained simulations, the Langevin thermostat was applied. In this revised manuscript, we have corrected and clarified the description of both temperature and pressure coupling.

3) Looking at the all-atom structure of the RNA (input files provided at the link) I found the RNA to be completely unstructured. This is an important assumption that needs to be discussed: that the authors (i) model the RNA as a random polymer and (ii) the RNA is unstructured, i.e. devoid of any secondary structure to begin with. Both assumptions require a reference. Looking at the trajectory, it was also very difficult to convince myself that Watson-Crick base-pairing was spontaneously observed in the pgRNA genome, something I was curious having seen the unstructured RNA at the beginning of the simulation. Could the authors pin-point to specific residues that validate the observation of RNA base-pairing? This is an important result that has a section on its own (Figure 6). How exactly are base-pairs defined would be an important information to mention in the methods section. This needs to be done for both all-atom and coarse-grained (CG) trajectory analysis (see line 192). The definition of base pairs in the CG model may overestimate true Watson-Crick base pairing.

Reply:

We sincerely appreciate the reviewer's valuable comments and suggestions. In this study, the wild-type pgRNA sequence was used in the all-atom MD simulations, enabling more realistic modeling of the inter-nucleotide interactions, particularly the Watson-Crick base-pairing. The initial pgRNA conformation was chosen to be unstructured, allowing us to more clearly identify base pairs that formed spontaneously during the simulations within the HBV capsid.

Following the reviewer's suggestion, we conducted additional all-atom MD simulations with different initial structures and performed more in-depth analyses (Fig. R1). First, we conducted two sets of new all-atom MD simulations, in which the

unstructured initial conformation and the structured initial conformation with rich secondary structure are used, respectively. The structured initial conformation was modelled by using AlphaFold3. In both sets of all-atom MD simulations, we observed the molecular events of canonical Watson-Crick base-pairing, although it is still unlikely to obtain converged sampling of pgRNA secondary structures due to timescale limit. To illustrate these processes, we have added new figures and video showing the time course of RNA base-pair formation for a representative nucleotide pair (Fig. R1a). Second, we extended our analyses of the coarse-grained simulation trajectories and presented an additional figure and video that more specifically highlight the base-pairing processes observed in the simulations (Fig. R1b,c). We also highlighted the specific sequence segments forming the dsRNA in Fig.6a, Fig. S16 and Fig. S17 in this revised manuscript. Together, the results of these new simulations and analyses further corroborate our observation that Watson-Crick base-pairing can occur within the HBV capsid.

In all-atom MD simulations, a pair of nucleotides was considered to form a base pair when the bases of two complementary nucleotides established hydrogen bonds, defined by a donor–acceptor heavy atom distance of less than 3.5 Å and an N–H–O or N–H–N angle greater than 120°. In the coarse-grained simulations, base-pair formation was defined when the distance of the base beads of two complementary nucleotides (with a sequence separation greater than 1) was less than a cutoff value of 6.5 Å. In this revised manuscript, we have incorporated these new results and discussions. The definition of base-pair formation has been explicitly added to the Methods section to improve clarity and reproducibility.

Figure R1. Dynamics of base pairs in MD simulations. (a, b) Snapshots of representative base-pairing events from all-atom (a) and coarse-grained (CG) simulations (b). (c) Representative folding and unfolding transitions of the cis-acting element ϵ observed in CG simulations. (d) Representative folding event of packaging site, PS3. (e) Time evolution of the number of dsRNA segments from CG simulations. (f) Lifetimes of dsRNA segments with lengths (L_{dsRNA}) ≤ 10 and > 10 . The lifetimes were estimated from a maximum-likelihood method.

Corresponding changes:

1. (Supplementary Material and Figure 6) We have added Fig. S17 and Movies 10–13 to illustrate the base-pairing events observed in both all-atom and coarse-grained simulations. We also highlighted the specific sequence segments forming the dsRNA in Fig. 6a, Fig. S16 and Fig. S17.

2. (page 12, line 254) The following sentence was added in this revised manuscript to describe the base pair dynamics:

“Typical processes involving the formation of base-pairing interactions are presented in Fig. S17 and Supplementary Movies 10–13.”

3. (page 12, line 263) The following sentences were added in this revised manuscript to describe the base pair dynamics:

“The formed base-pairs exhibit pronounced dynamics in the CG simulations. Unwinding events of dsRNA can be clearly observed in the simulation trajectories (Fig. S17b and Supplementary Movie 11). Moreover, substantial variation in the number of formed dsRNA segments during the CG MD simulations supports the highly dynamic nature of the dsRNA conformation (Fig. S17e).”

4. (page 16, line 445) The following sentences were added in the Methods section to clarify the definition of base pairs for all-atom MD simulations:

“To identify Watson-Crick base pairs, we applied geometric criteria to complementary nucleotides with a sequence separation greater than one. A pair of nucleotides was considered to form a base pair when the bases of two complementary nucleotides established hydrogen bonds, defined by a donor–acceptor heavy atom distance of less than 3.5 Å and the N–H–O or N–H–N angle exceeded 120°.”

5. (page 17, line 475) The following sentence was added in the Methods section to clarify the definition of base pairs for coarse-grained MD simulations:

“Watson-Crick base-pair was assigned when the distance between the base beads of

two complementary nucleotides (with a sequence separation greater than one) was less than a cutoff value of 6.5 Å.”

4) Line 70: The “pgRNA quickly condensed along the inner surface of the capsid”:
Could this be because of bad initial configuration of the RNA whereby any internal stabilization via base-pairing is completely dominated by strong nucleic acid-Arginine interactions? A movie of the initial spherical confinement to this shell-like arrangement would be a useful addition.

Reply:

We thank the reviewer for the helpful comment. In this revised manuscript, we have included new MD simulations initiated from pgRNA structures with predicted secondary structures, using both all-atom and coarse-grained models. Consistent with our earlier simulations starting from unstructured pgRNA conformation, these new simulation results show that pgRNA rapidly condenses along the inner surface of the capsid, forming a shell-like arrangement (Fig. R2). These results indicate that pgRNA condensation along the capsid interior is a robust process, largely insensitive to the choice of initial pgRNA structure. Following the reviewer’s suggestion, we prepared new movies and figures to more clearly illustrate the formation process of the shell-like conformation of pgRNA. In this revised manuscript, we have added the above results and discussion.

Figure R2. Conformations of pgRNA from the MD simulations with different initial structures. (a,b) The starting and final configurations (a), and the corresponding radial distribution functions of NTD, CTD, and pgRNA in the final configuration (b), from the all-atom MD simulation initiated from an unstructured pgRNA. (c,d) Same as (a,b), but for the all-atom MD simulation initiated from a pgRNA structure predicted by AlphaFold3. (e,f) Same as (a,b) but for the CG MD simulation initiated from a pgRNA structure predicted by AlphaFold3.

Corresponding changes:

1. (Supplementary Material) We have added Fig. S1 and Movies 2-4 from the MD simulations with different initial structures to illustrate that the results are insensitive to the initial structures.

2. (page 3, line 79) The following paragraph was added in the main text of this revised manuscript to describe the results of all-atom MD simulations with different initial structures:

“To further assess the dependence of the results on the initial structure and force fields, we conducted two additional sets of all-atom MD simulations initiated from pgRNA structures without pre-assigned secondary structures and with predicted secondary structures, respectively. Both simulations employed an alternative force field (amber14sb_OL15 force field with NBFIX correction)^{35, 36}. Consistent with the above result (Fig. 1d), these simulations also revealed rapid condensation of pgRNA along the inner surface of the capsid, forming a shell-like arrangement (Fig. S1a,b and Supplementary Movies 2-3). These results indicate that the observed pgRNA condensation is a robust process, largely insensitive to the choice of initial pgRNA structures and force fields.”

3. (page 5, line) The following sentences were added in the main text of this revised manuscript to describe the results of CG MD simulations with different initial structures:

“We also conducted CG simulations initiated from pgRNA structures with predicted secondary structures, and comparable results were observed (Fig. S1c and Supplementary Movie 4).”

5) Traditionally, MD force fields can overestimate interactions between highly charged moieties, that includes Arginine residues binding strongly to the phosphate groups on nucleic acids. Conventional practice prevails the use of specific NBFIXs or charge rescaling to tackle this, none that were used in this study. Could the authors comment on the choice of force field, whether it can adequately represent these interactions. This issue is particularly critical because the pgRNA-CTD condensate formation is driven by electrostatics, and overestimating these interactions could result in an artificial state that mimics condensate properties. Nonetheless, the slab simulations

provide a robust representation.

Reply:

In this work, the ff14sb and χ_{OL3} force fields were used for the protein and RNA, respectively. Following the reviewer's comment, we added additional all-atom MD simulations employing an Amber force field (amber14sb_OL15) with NBFIX correction to assess the force field dependence of the results. The key observations, such as the formation of shell-like RNA arrangement and RNA base-pairing, were consistently reproduced in these new simulations, indicating that the observed condensate properties are robust (Figs. R1a and R2). In this revised manuscript, we have added the above results and discussions.

Corresponding changes:

1. (page 16, line 431) *The following sentence was added in the Methods section of this revised manuscript:*

“To assess the force field dependence of the results, we performed additional all-atom MD simulations employing the amber14sb_OL15 force field with NBFIX correction.^{35, 36}”

2. (page 3, line 79) The following paragraph was added in the main text of this revised manuscript to describe the results of all-atom MD simulations with different initial structures (The same change as in the reply of comments 4):

“To further assess the dependence of the results on the initial structure and force fields, we conducted two additional sets of all-atom MD simulations initiated from pgRNA structures without pre-assigned secondary structures and with predicted secondary structures, respectively. Both simulations employed an alternative force field (amber14sb_OL15 force field with NBFIX correction)^{35, 36}. Consistent with the above result (Fig. 1d), these simulations also revealed rapid condensation of pgRNA along the inner surface of the capsid, forming a shell-like arrangement (Fig. S1a,b and Supplementary Movies 2-3). These results indicate that the observed pgRNA condensation is a robust process, largely insensitive to the choice of initial pgRNA structures and force fields.”

6) Line 114: Using only a single 50nt segment out of the 3200nts is questionable as “randomly selected”. Also Fig. S2b’s caption mentions this segment as nucleotides 1-50. If so, the “randomly selected” in line 115 should be removed. The RNA is confined and one may argue that the true MSDs can be very sensitive to the choice of the fragment, in particular due to the observation of both voids and high density regions. The same analysis should therefore be based on an average over several such fragments. Why can’t the analysis be done for all such possible fragments?

Reply:

We thank the reviewer for the helpful comments and suggestion. We agree that calculating the average MSD across multiple RNA segments provides a more appropriate characterization of diffusion ability. Accordingly, we divided the pgRNA chain into segments of 100 nucleotides and computed the average MSD among these segments (Figs. R3, R4). Consistent with our original results, the average MSD indicates significant diffusion of pgRNA along the shell-like condensate. In this revised manuscript, we updated the corresponding figures and refined the related description of the analysis method to reflect this improvement.

Figure R3. Mean-squared displacement (MSD) of pgRNA, averaged over different 100-nt segments of the full-length pgRNA, under varying salt conditions.

Figure R4. Mean-squared displacement (MSD) of pgRNA, averaged over different 100-nt segments of the full-length pgRNA, under varying temperatures.

Corresponding changes:

1. (Main text figure and Supplementary Material) The Fig.2k and Fig. S2d were updated in this revised manuscript.

2. (page 5, line 137) The following sentences in the main text of this revised manuscript were refined to reflect the MSD results with updated computing method:

“By dividing the pgRNA sequence into 100-nucleotide segments, we also calculated the mean square displacement (MSD) averaged among these segments from the aforementioned CG MD simulations. The MSD results reveal substantial diffusivity (Figs. 2k and S4d), particularly under physiological conditions (Supplementary Movie 5).”

7) Line 174-177: The authors mention that the RNA density is much lower at the threefold symmetry sites, compared to fivefold and sixfold. This is an interesting result. Could the authors comment on why this may be so? Also, it is not clear whether the analysis done in Figure 5 used only the all-atom trajectories or does it also include coarse-grained trajectories. Accordingly the caption needs to be updated, that only says “MD simulations.”

Reply:

We thank the reviewer for the helpful comment and suggestion. Due to the co-condensation of CTDs and pgRNA, the density distribution of pgRNA is influenced by the spatial locations of CTDs along the inner surface of the capsid. Based on the icosahedron arrangement of the core proteins, CTDs are more densely localized near the fivefold and sixfold symmetry axes (Fig. R5). Consequently, pgRNA density tends to be enriched around these symmetry sites, reflecting the underlying organization of the capsid proteins.

In this study, the RNA density map and its distribution were estimated from the coarse-grained trajectories. In this revised manuscript, we have added the related discussion on the pgRNA distribution. The caption of Figure 5 has also been modified accordingly to improve clarity.

Figure R5. Density of CG beads at different symmetry sites estimated from CG simulations.

Corresponding changes:

1. (Supplementary Material) We have added Fig. S15 to demonstrate the distribution of CTDs across different symmetry sites.

2. (page 9, line 235) The following sentences were added in the main text of this revised manuscript:

“The enrichment of pgRNA around the fivefold and sixfold symmetry sites is attributed to the co-condensation of CTDs and pgRNA. As CTDs are relatively more concentrated at the fivefold and sixfold symmetry sites on the inner surface (Fig. S15), the average density of pgRNA at these sites is consequently higher.”

3. (Figure 5 caption) The Figure 5 caption was updated to clarify that the results were obtained from CG MD simulations.

8) Line 193-195: The authors argue that a marked increase in the number of base pairs indicates that LLPS enhances the pgRNA secondary structure. Could the increase of base pairs be just due to a bad initial structure of the pgRNA that is devoid of secondary structure at the beginning? If so, that LLPS enhances this formation would be purely speculative.

Reply:

We thank the reviewer for the constructive comments. To further validate the observation that LLPS enhances base-pairing, we conducted two additional sets of coarse-grained simulations with the initial structure of the capsid-confined pgRNA possessing secondary structures. In the first set of simulations, electrostatic interactions between RNA and the CTDs of capsid proteins were turned off, thereby preventing phase separation. Under this condition, the number of base pairs (N_{bp}) have no significant changes during the simulations (Fig. R6). In the second set of simulations, electrostatic interactions between RNA and CTDs were retained. In this case, N_{bp} increased continuously over the time course of $\sim 1 \times 10^7$ MD steps (Fig. R6). These results validate that LLPS promotes base-pair formation, and that this effect is robust and insensitive to the choice of initial pgRNA structure. For each condition, two independent simulations were conducted. In this revised manuscript, we have added the above results and discussions to more clearly illustrate the effect of LLPS on the formation of pgRNA base pairs.

Figure R6. Temporal evolution of the number of base pairs (N_{bp}) from CG simulations of the pgRNA-containing HBV capsid under conditions with and without the occurrence of phase separation, starting from a pgRNA conformation with predicted secondary structures. Two independent simulations were conducted for each case.

Corresponding changes:

1. (Supplementary Material) We have added Fig. S18 to demonstrate that the observed role of LLPS in prompting base pairing is independent of the initial pgRNA conformations.
2. (page 12, line 257) The following sentence was added in the main text of this revised manuscript.

“We also conducted additional CG simulations with the initial structure of the capsid-confined pgRNA possessing pre-assigned secondary structures, and similar role of LLPS in promoting base-pair formation was observed (Fig. S18).”

9) How is long-range base pairing defined? This is hard to convince from nucleotide contacts plots (Fig. 6j) because nucleotides distant w.r.t. their index/residue may still be spatially located in relative proximity and vice versa. Accordingly, the text may need to be updated (for e.g. line 215). The use of the word “significant” on line 235 is speculative.

Reply:

We thank the reviewer for the comment and suggestion. In this work, the long-range base pairing is defined on the basis of the sequence separation, sequence complementarity, and distance. When the distance between the base beads of two complementary nucleotides in the structures of the coarse-grained simulations is less than a cutoff value (6.5 Å), they form a base pair. The long-range base pairs refer to those formed by nucleotides with large “sequence separation”. In this study, base pairs with sequence separation larger than 100 were defined as long-range base pairs. We calculated average number of long-range base pairs for the simulations with and without phase separation. The results show that the number of long-range base pairs increases from 31 to 787 with the introduction of electrostatic interactions between pgRNA and CTDs, suggesting that LLPS favors long-range base-pairing (Fig. R7). The number of long-range base pairs is also higher than that of the predicted pgRNA structure (410). In this revised manuscript, we updated the related text to more accurately describe the results, and the word “significant” was rephrased.

Figure R7. Number of base pairs (N_{bp}) formed by long-range nucleotides (with sequence separation greater than 100) from CG simulations of the pgRNA-containing HBV capsid under conditions with and without the occurrence of LLPS.

Corresponding changes:

1. (Supplementary Material) We have added Fig. S23 to quantify the role of phase separation in prompting long-range base pairing.
2. (page 13, line 335) The following sentences were added in the main text of this revised manuscript:

“Here, we can observe an increase of long-range base pairing (Fig. 7d and Fig. S23), which is essential for enabling the template switches as discussed above. In this study, the base pairs with sequence separation larger than 100 are classified as long-range base pairs.”

Minor concerns:

1) Line 17: I would recommend citing Bloomfield, V. A. DNA condensation by multivalent cations. Biopolymers 44, 269–282 (1997), as the first to suggest “significant electrostatic repulsions”.

Reply and corresponding changes:

We thank the reviewer for bringing this important paper into our attention. In this revised manuscript, we added the citation of this paper.

2) Figure 2c-f: The colors are not defined in the caption.

Reply and corresponding changes:

In Figure 2c-f, the NTDs, CTDs, and pgRNA are depicted in gray, blue and wheat, respectively. In this revised manuscript, we added the color definition for Figure 2c-f in

the caption.

3) Figure 4a: It is not clear how the anchored CTD is modeled. Is it all the 37 residues of the CTD? From the image, the anchored NTD is not fully modeled and it is unclear where the anchor point is. This should be updated in line 152 (“N-termini fixed”).

Reply and corresponding changes:

In the simulations of the two-dimensional system, the N-terminal domain of capsid protein was omitted, and the peptides corresponding to CTDs (residues 150-185) were anchored to a two-dimensional plane by applying a harmonic potential between the N-terminal beads of CTDs and the reference positions in the two-dimensional plane. The reference positions are uniformly distributed with the density roughly equivalent to that of the CTDs in the capsid. In this revised manuscript, we updated the related description to improve clarity.

1. (page 7, line 175) The following sentences were refined in this revised manuscript to clarify the protocol for the construction of the two-dimensional system:

“In this two-dimensional system, the NTD of the capsid protein was omitted, and the N-terminus of the peptides corresponding to CTDs were anchored to a two-dimensional plane by applying a harmonic potential. The reference positions were uniformly distributed at a density approximately equivalent to that of the CTDs in the native capsid.”

4) Figure 5a: The isosurface density scale is missing. From the image it may falsely appear that there are no RNA residues in the threefold symmetry sites, whereas this is likely only due to the isosurface density used to create the image.

Reply and corresponding changes:

We thank the reviewer for the comments and suggestion. In this revised manuscript, we added the isosurface density scale in Figure 5a. To improve clarity, we also emphasized that the density less than the lower-limit of the density scale is not plotted.

5) Figure 5c: Color scale is missing.

Reply and corresponding changes:

In this revised manuscript, we added the color scale in Figure 5c.

6) Line 199-203: Are these simulations all-atom or coarse grained?

Reply and corresponding changes:

These simulations were conducted with the coarse-grained models. In this revised manuscript, we have refined the related text to explicitly describe the simulation method and improve clarity.

7) The MSD plots (Figure 2k and 7e) suggest the diffusion is very much Brownian like (straight lines at longer lag times). This is a bit counter-intuitive at longer lag times because of the capsid confinement.

Reply and corresponding changes:

In Figure 2k and 7e, the lag-time scale of the diffusion is limited to 1.5×10^7 MD step, during which the capsid confinement is less important. When the lag-time scale of the diffusion becomes longer, the slope of the MSD plot gets smaller because of the capsid confinement (Fig. R8). In this revised manuscript, we have added Fig. S5 to illustrate the diffusion behavior at longer lag time (3×10^7 MD step) and the effect of capsid confinement.

Figure R8. Mean squared displacement (MSD) of pgRNA, averaged over different 100-nt segments of the full-length pgRNA from CG simulations at temperature of 300K and salt concentration of 150 mM. The slope of the MSD plot gets smaller at longer lag-time because of the capsid confinement.

8) Line 92: The “an” should be “a”.

Reply and corresponding changes:

We thank the reviewer for pointing out the grammatical error. In this revised manuscript, we corrected this grammatical error.

Reviewer #2 (Remarks to the Author):

The manuscript describes results of a detailed study, mostly by means of multiscale molecular dynamics computer simulations and partly by experimentation, of the structural organization and condensation of the pregenomic RNA in the capsid of the virus HBV. The authors focus on the role of the positively charged, structurally disordered C-terminal RNA binding domain of the coat protein, often called Arginine Rich Motif or ARM, in relation with their spatial confinement in the capsid. Parenthetically, in the manuscript the ARMs are called CTD, short for C terminal domain.

The main point the authors try to convince the readers of is that the structural arrangement of the RNA in HBV is driven by phase separation, arguing that viruses exploit what is known as liquid-liquid phase separation or LLPS. LLPS has attracted a lot of attention in biophysical literature recently. Often (but not always) driven by interaction of, e.g., RNAs with positively charged, structurally disordered proteins, and I understand why seeing a correspondence between virus assembly and LLPS in biology is such an attractive one.

I find the paper hugely interesting, and I am particularly impressed by the comprehensiveness of the simulation study. The findings are highly significant, and, if accurate, would further our understanding of the structure of viruses and appeal to a broad readership. Also, the paper is very well written, well-structured and easy to read. I do find pushing the LLPS narrative so forcefully as is done in the manuscript at times a little too much. As a matter of fact, I am not yet convinced that the case made by the authors actually hold water, irrespective of the interesting findings they present.

Reply:

We thank the reviewer for the kind comments.

Some might view this a semantic issue, but in my view absorption of what in the end is a polymer (the ssRNA) into a structured polymer brush (the ARMs emanating into the cavity of a capsid) is not the same as macroscopic phase separation, in the same way that condensation of a gas in bulk is not the same as condensation of a gas onto an adsorbing surface. Strictly speaking, the assembly of coat proteins around an RNA cannot represent a true phase transition on account of its finite size, noting that the droplet phases typically found in LLPS must be transient. The physics of adsorption of RNA onto immobilized ARMs part of a capsid is very different from the co-condensation of ARMs with RNA in free solution – the authors are aware of this but seem to downplay this. (polymer brush phase separation)

Reply:

We fully agree with the reviewer's view that the condensation of pgRNA with CTDs

inside the HBV capsid is very different from the macroscopic LLPS in free solution. First, the CTD is anchored to the inner surface of capsid, and therefore it cannot diffuse freely to form larger droplet by co-condensation with pgRNA. Second, as the reviewer noted, the finite size of the system makes the condensate structure distinct from bulk-phase LLPS, where low- and high-density phase are well separated to form round-shaped droplets.

At the same time, this CTD-RNA co-condensation shares some important features with typical macroscopic LLPS in free solution: 1) coexistence of high- and low-density regions; 2) liquid-like dynamics as shown by the high molecular mobility of condensate and cluster coalescence events; and 3) high sensitivity to perturbation in temperature and salt concentration. Inspired by the reviewer's comments, we emphasized these differences and similarities in this revised manuscript, highlighting how CTD-pgRNA co-condensation within the HBV capsid contrasts with LLPS in free solution.

Inspired by the reviewer's comments, we carefully surveyed the physical studies on polymer brush. Interestingly, numerous interesting computational and experimental works have examined the electrostatic regulation of the phase behavior of polyelectrolyte brush (*Macromolecules* 2009, 42, 7194-7202; *PNAS*, 2010, 107:5300-5; *Sci. Adv.* 2017, 3: eaao1497). Previous studies have shown that polyelectrolyte brush can undergo morphological transition by modulating electrostatic interactions, for example, through the absorption of multivalent ions (*Sci. Adv.* 2017, 3: eaao1497) or by pH-dependent modulation of the charge states of charged groups (*PNAS*, 2010, 107:5300-5). Such electrostatically regulated morphology transition of uniformed distributed polyelectrolyte brushes is often referred to as "lateral phase separation" or "planar phase separation" (*Biointerphases*, 2022, 17:060802; *J. Polym. Sci. B.* 2017, 55: 370–377). Compared to typical polymer brush system, RNA in the HBV capsid is not only a participant in the phase behavior as an electrostatic regulator, but also a central component of the condensates. Notably, a previous NMR study revealed that the HBV-RNA interactions within the capsid are highly fluidity (*JACS*, 2022, 144:8536-8550.), in line with the liquid-like dynamics of the CTD-RNA condensate formed by LLPS as discussed in this work. In this revised manuscript, we have extended the discussion to relate the organization of the HBV pgRNA genome to the phase separation of polymer brush described in the polymer physics literature. Furthermore, we have added experiment data and supplementary all-atom simulations to support the morphology transition observed in the two-dimensional CTD-RNA system.

Corresponding changes:

1. (page 15, line 371) The following paragraph was added in this revised manuscript to emphasize the difference between the phase separation of the CTD-RNA system from the macroscopic phase separation in free solution:

"It is also worth emphasizing that the above discussed phase separation of the pgRNA

genome within the HBV capsid differs from the macroscopic LLPS in free solution in the following two important aspects: i) the CTD is anchored to the inner surface of capsid and therefore cannot diffuse freely to form larger droplets through co-condensation with pgRNA; and ii) the finite size of the system makes the condensate structure distinct from bulk-phase LLPS, where low- and high-density phases are typically well separated to form round-shaped droplets. At the same time, this CTD-RNA co-condensation shares key features with typical macroscopic LLPS, including the coexistence of high- and low-density regions, liquid-like condensate dynamics, and sensitivity to temperature and salt concentrations.”

2. (page 9, line 210) The following sentences were added in this revised manuscript to discuss the similarity between the phase separation of the two-dimensional CTD-RNA system and the lateral phase separation observed in polymer brush:

“The two-dimensional planar system described above is analogous to a polymer brush. Previous studies have showed that polyelectrolytes grafted onto a two-dimensional substrate can segregate into coexisting domains in response to electrostatic regulation^{61–63}, a phenomenon often referred to as “lateral phase separation”⁶⁴ or “planar phase separation”⁶⁵. Compared to typical polyelectrolyte brush system, RNA in the HBV capsid is not only a participant in the phase behavior as an electrostatic regulator, but also a central component of the condensates.

There are a few other issues that I have with the manuscript that need to be addressed before I would advise in favour of publication.

1. There is a lot of theoretical and simulation literature on the structure of (model) RNAs in virus capsids, and the readers should be aware of that. I am referring to, e.g., the work of Hagan, Zandi and Podgornik to name but a few. Of particular importance for the topic of the manuscript is the work of Hagan in eLife 2013; 2: e00632, and Zandi and Podgornik in Phys. Rev. E 2014; 89: 032707 and Zandi in Phys. Rev. E 2020; 102: 062423. These, and other works, highlight the importance of the localization of the positive charges, but also the configurational (base-pairing) response of the RNA to the binding to the ARMs. See also the review paper of Zandi and collaborators, Physics Reports 2020; 847: 1–102.

Reply:

We thank the reviewer for bringing these interesting papers to our attention. These works clearly demonstrate the close interplay between the secondary structural organization of RNA, the localization of positive charges within the arginine-rich motifs of capsid proteins, and the optimal length of encapsulated RNA genome, thereby highlighting the importance of physics-based approaches in elucidating virion

organization. In this revised manuscript, we have incorporated a discussion of these previous studies to better contextualize our findings.

Corresponding changes:

1. (page 14, line 393) The following sentences were added to discuss the existing theoretical studies of virial assembly:

“The key role of phase separation in the structural organization of HBV pgRNA genome demonstrated in this work underscores the importance of physical principles in elucidating the viral life cycle. Indeed, substantial advances in understanding viral structure and assembly have been achieved through physics-based methodologies, which have successfully explained a wide range of experimental observations.⁹⁶ For instance, by using a mean-field theory, Zandi and coworkers have illustrated the effects of charge distribution of capsid protein and RNA topology on the amount of packaged RNA based on the genome configurational entropy and electrostatics.^{97, 98} Base-pairing in RNA can trigger a conformational transition of RNA genome from an extended coil to a compact globule.⁹⁹ In addition, kinetic and thermodynamic analysis of viral capsid assembly by Hagan et al revealed the importance of genome tertiary structure in capsid assembly and elucidated the key factors contributing to overcharging effect.⁷³ Collectively, these studies highlight the close interplay between RNA secondary structure, the electrostatic regulation, and the optimal length of the encapsulated genome, emphasizing the value of physics-based approaches in elucidating virion organization.”

2. I do not believe that the observed coexistence of high- and low-density regions in the absorption layer near the ARMs indicates phase separation, as is implied on page 3, lines 77 and 78. The difference in RNA density in the cavity and that in the absorption layer would be in closer to correspondence to complex coacervation of ARMs and RNA, discussed also in the main text. I am very skeptical about the findings from the simulations that phase separation takes place in the absorption layer in the virus (fig. 2), and on the flat surface with end-grafted ARMs (fig. 4) unless the salt concentration is sufficiently high. The reason is that this leaves the regions with low RNA density with nearly naked ARMs, so without the benefit of attractive interaction with the RNA whilst the higher-density regions must suffer from higher self-repulsion of the RNA. No experimental verification has been done, as was with the bulk phase separation of RNAs and ARMs (fig. 3). I would strongly recommend experimental verification, knowing full well that this is a major endeavor. The discovery of dewetting of a polyanion such as RNA adsorbed on a polycationic surface would represent a major surprise in the field.

Reply:

We thank the reviewer for the insightful comments and suggestion. By carefully examining the structural features of the high- and low-density regions in the absorption

layer, we found that although one terminus of the CTD is uniformly grafted to the planar surface, the overall CTD density distribution is highly heterogeneous. Typically, the free-ends of the CTD chains tend to orientate from low density regions toward high density regions (Fig. R9a,b), leading to the formation of nanometer-sized clusters. As a result, high-density regions are enriched in both CTD and RNA, whereas low-density regions contain relatively few CTD moieties due to covalent anchoring (Fig. R9a,b). In the high-density regions, close contacts between positively charged CTD segments and negatively charged RNA segments promote favorable electrostatic attraction. Meanwhile, in the low-density regions, the larger inter-CTD distances (due to the outward orientation) reduce unfavorable electrostatic repulsion between CTD segments. Thus, RNA dewetting is facilitated by the enhanced electrostatic attraction in high-density regions and the reduced electrostatic repulsion in low-density regions.

We agree with the reviewer's comment that when low-density regions are instead enriched in naked CTD segments, electrostatic interactions would become unfavorable for phase separation. This scenario arises when the CTD density is much higher than the conditions studied in this work. Indeed, in newly added simulations with elevated CTD densities, phase separation gradually becomes invisible by increasing CTD density (Fig. R9c-f), likely due to unfavorable electrostatic repulsion among CTD segments in the low-density region when RNA is segregated into clusters, as commented by the reviewer.

Encouraged by the reviewer's suggestions, we tried to perform new experiments to further support the CTD-RNA microphase separation in the two-dimensional system. Because CTD segments are short (36 amino acid) and the resulting clusters are typically nanometer in size, which is too small to be detected by fluorescence microscope or atomic force microscope (AFM), we adopted an alternative strategy. Specifically, we grafted PEG onto a substrate and decorated the free PEG terminus with CTD segments. This extended the effective polymer length of the two-dimensional brush, enabling the formation of larger clusters upon RNA addition. We then employed AFM to examine the morphological changes induced by adding RNA to the constructed PEG-CTD layer. As expected, the AFM measurements revealed that RNA addition promotes morphological heterogeneity in the hybrid PEG-CTD layer (Fig. R10), resembling the lateral phase separation observed in early studies (Sci. Adv. 2017, 3: eaao1497). Because AFM morphology has limited resolution and it is difficult to absolutely rule out alternative origins for the observed morphological transition beyond phase separation, we also carried out additional all-atom MD simulations of the two-dimensional CTD layer with RNA segments (Figs. R11a,b). Consistent with the coarse-grained MD simulations, these all-atom MD simulations demonstrated RNA dewetting in the two-dimensional system, although the simulation length is limited due to computational demands.

In this revised manuscript, we have incorporated these new results and discussions to more clearly elucidate the molecular mechanism underlying two-dimensional phase separation.

Figure R9. Effect of CTD density on the CTD-RNA condensation in the two-dimensional system. (a,b) Representative snapshots for the CG simulations with the addition of pgRNA (a) and A₁₅ RNA (b) to the two-dimensional system with the native CTD density at the salt concentration of 50mM. (c,d) Same as (a,b) but at the salt concentration of 150mM. (e,f) Same as (a,b) but with increased CTD density at the salt concentration of 150mM. d_N refers to the distance between the N-terminal anchoring beads of neighboring CTDs.

Figure R10. AFM characterization of RNA induced morphological transition of PEG-CTD layer. (a,b) AFM tapping mode height images of PEG-CTD layer before (a) and after (b) the addition of A₁₅ RNA. (c,d) AFM phase images of PEG-CTD layer before (c) and after (d) the addition of A₁₅ RNA.

Figure R11. All-atom MD simulations of phase separation of the two-dimensional CTD-RNA system. (a) Representative snapshot of all-atom MD simulation for the two-dimensional system containing 15×16 array of CTD chains with the simulation length of 100 ns. (b) Representative snapshot of all-atom MD simulation for the two-dimensional system containing 10×10 array of CTD chains with the simulation length of 200 ns. (c,d) Distribution of Na⁺ (c) and Cl⁻ (d) across the two-dimensional layer for the two-dimensional system containing 15×16 array of CTD chains. (e) Fraction of ions within the high- and low-density regions of CTD-RNA clusters.

Corresponding changes:

- (Supplementary Material) We have added Fig. S11-14 to illustrate the results of coarse-grained MD simulations, AFM characterization, and all-atom MD simulations of the phase separation for a two-dimensional system.
- (page 9, line 198) The following paragraph was added to discuss the physical basis of the phase separation for the two-dimensional CTD-RNA system.:

“Structural analysis of the high- and low-density regions in the two-dimensional condensate layer reveals that, although one terminus of each CTD is uniformly grafted to a planar surface, the overall CTD density distribution is highly heterogeneous. Typically, the free ends of CTD chains tend to orientate from low density regions toward high density regions (Fig. S11a,b), leading to the formation of nanometer-sized clusters. Consequently, high-density regions are enriched in both CTD and RNA, whereas low-density regions contain relatively fewer CTD moieties due to covalent anchoring. In the high-density regions, close contacts between positively charged CTD segments and negatively charged RNA segments promote favorable electrostatic attractions. Meanwhile, the larger inter-CTD distances (due to outward orientation) in low-density regions reduce unfavorable electrostatic repulsion. Thus, RNA dewetting is facilitated by the enhanced electrostatic attraction in high-density regions and the reduced

electrostatic repulsion in low-density regions. Further investigations established that the two-dimensional phase separation of the CTD-RNA mixture system depends critically on the surface density of CTDs. Altering CTD density by changing inter-chain distances under physiological conditions diminished phase separation (Figs. S11, S12), likely due to alteration of the electrostatic balance.”

3. (page 9, line 214) The following sentences were added to describe the AFM experiment and all-atom MD simulation results of the phase separation of the two-dimensional system:

“To further support the two-dimensional phase separation of the CTD–RNA system, we employed atomic force microscope (AFM) to characterize the morphological transition. Because of the short length of CTD peptides, the resulting condensate are typically nanometer in size, which is too small to be detected by fluorescence microscope or AFM. To overcome this limitation, we grafted polyethylene glycol (PEG) onto a planar substrate and decorated the free PEG terminus with CTD segments. This extended the effective polymer length, enabling the formation of larger clusters upon RNA addition. AFM measurements revealed that RNA addition promotes morphological heterogeneity in the two-dimensional system (Fig. S13), resembling the lateral phase separation observed in early studies⁶¹. Because AFM morphology has limited resolution and it is difficult to absolutely rule out alternative origins for the observed morphological transition beyond phase separation, we also carried out additional all-atom MD simulations of the two-dimensional CTD layer mixed with RNA segments. These all-atom MD simulations clearly demonstrated RNA dewetting (Fig S14a,b).”

3. The in-layer phase separation shown in fig. 4 was studied for different ionic strengths, but only a single RNA length. It would be useful to investigate how the length of the RNA influences this phase separation. Also, the phase separation, if real, should lead to inhomogeneous distribution of mobile ions. Not much is said about this in the manuscript. Note also, that if phase separation indeed takes place, the high- and low-density regions should increase in size in time and coarsen, e.g., by coalescence. Does coarsening take place?

Reply:

In this revised manuscript, we have added a series of new coarse-grained simulations with much shorter RNA segments, specifically A₁₅ RNA segments (Fig. R12). Similar phase-separation behaviors were observed under varying salt concentrations, phase separation becoming more pronounced at lower salt concentrations due to stronger electrostatic interactions. Following the reviewer’s comments, we also analyzed the coarsening dynamics in these simulations and identified clear coalescence events (Fig. R13). Because one terminus of each CTD is anchored, the sizes of the high-density clusters remain limited and continuous coalescence of smaller clusters into larger ones becomes difficult.

Following the reviewer's comments, we also characterized the distribution of mobile ions. Since the influence of salts is implicitly accounted for through the Debye-Hückel model in the coarse-grained simulations, we performed additional analysis using all-atom MD simulations to examine whether ions exhibit inhomogeneous distribution (Figs. R11c-e and R14d,e). The results revealed that the distributions of Na⁺ and Cl⁻ around the condensate layer are highly heterogeneous and closely correlated with the spatial organization of the CTD-RNA condensate (Fig. R11e), owing to the greater accumulation of ions in high-density CTD-RNA regions.

In this revised manuscript, we have added the above new results and discussion to illustrate the in-layer phase separation for shorter RNA segments, mobile ion distributions, and the coarsening of CTD-RNA condensate.

Figure R12. Representative simulation snapshots from CG simulations of the two-dimensional system at varying salt concentrations for A₁₅ RNA molecules along the CTD layer.

Figure R13. Fusion events captured in CG simulations of A₁₅ RNA molecules along the two-dimensional CTD layer at the salt concentration of 50 mM.

Figure R14. Distribution of mobile ions estimated from the all-atom MD simulations. (a) Ratio of the cumulative number of Na^+ (red) and Cl^- (blue) ions flowing into and out of the capsid over successive 5 ns intervals. (b) Radial distribution functions of mobile ions relative to the capsid center. For comparison, the distribution of NTD, CTD, and pgRNA were also shown. (c) Radial distribution functions of mobile ions relative to the capsid center calculated at three different simulation times (80ns, 90ns, and 100ns). (d,e) Representative two-dimensional density distributions of Na^+ (a) and Cl^- (b) plotted along spherical coordinates (θ , ϕ). The density distributions in (d,e) were calculated for the region indicated by two vertical lines in (b,c).

Corresponding changes:

1. (Supplementary Material) Fig. S2,S9-S10, and Supplementary Movie 9 were added.

2. (page 7, line 183) The following sentences were added in this revised manuscript:

“Similar phase-separation behaviors were observed when much shorter RNA segments (A15 RNA segments) were added (Fig. S9). Interestingly, we observed clear coarsening events, in which two small clusters coalesced into a larger cluster (Fig. S10 and Supplementary Movie 9), a hallmark feature of macroscopic phase separation. However, because one terminus of each CTD is anchored, the sizes of the high-density clusters remain limited and continuous coalescence into larger droplets becomes difficult.”

3. (page 9, line 224) The following sentence was added to describe the heterogeneous distributions of Na^+ and Cl^- in all-atom simulations:

“Correspondingly, the distributions of Na⁺ and Cl⁻ around the condensate layer are highly heterogeneous (Fig. S14c,d) and closely correlated with the spatial organization of the CTD-RNA condensate (Fig. S14e).”

4. The structure of the RNA in particular in relation to base pairing is discussed on page 9 and 10 (figure 6). There is quite some work on the structure of viral ssRNAs, and why they are different from random RNAs or cellular RNAs. See, e.g., the work of Gelbart and others on this topic in PLoS ONE 2014; 9: e105875 and Biophys. J. 2017; 113: 339-347, to cite a few of many papers these authors wrote on this topic. Also, theoretical work (e.g. of Zandi, see earlier) indicates that interaction with the ARMs changes the base pairing. It would be interesting to get more information on the structure of the RNA in the virus, such as maximum ladder distances, length distributions of hairpin structures, and so on. The authors also mention nematic alignment of clustered hairpins but do not elaborate on how this was established: by eye or by calculating the largest eigenvalue and associated eigenvector of the local Q tensor? This would also produce a director field configuration – in that case one would expect four half-integer disclination defects to appear. There is quite a bit of literature on this.

Reply:

We thank the reviewer for bring these interesting researches to our attention. During the revision of this manuscript, we carefully examined these highly relevant prior works and incorporated discussions comparing them with the findings of this study. In accordance with the reviewer’s suggestion, we have also included additional analyses of the structural features of the viral RNA, such as the maximum ladder distance (MLD), the length distributions of hairpin structures, and the scalar order parameter S.

For the dsRNA structures, we analyzed the length distributions at varying salt concentrations (Fig. R15a-c). The analysis shows that the dsRNA regions become shorter with increasing salt concentration. This observation suggests that the CTD-mediated structural organization of pgRNA can be modulated by ionic strength.

Following the reviewer’s suggestion, we computed the largest eigenvalue (S) of the local Q-tensor to characterize the nematic alignment of neighboring dsRNA strands. Specifically, S was evaluated for each spatially adjacent pair of dsRNA molecules and then averaged across all pairs. Across different salt concentrations, the average value of S ranged between 0.3 and 0.5, which is characteristic of systems in the nematic phase (Fig. R15d). As expected, the average S value decreased with increasing salt concentration, reflecting the enhanced flexibility of dsRNA under high-salt conditions.

In addition, our analysis revealed that the maximum ladder distance (MLD), which reflects the overall extension of RNA, was estimated to be between 110 and 120 under conditions where phase separation is prominent (Fig. R15e). This value is substantially

lower than the MLD of a randomly permuted single-stranded RNA of comparable sequence length (~300 for a 3000-nt RNA), consistent with previous findings that viral RNAs adopt more compact conformations than their randomized counterparts (PNAS, 2008, 105: 16153–16158; PLoS ONE, 2014, 9: e105875). Furthermore, we observed a pronounced decrease in MLD at a high salt concentration, where phase separation is absent. This is likely because the suppression of phase separation impedes base pairing as shown in Fig. 6c, thereby leading to a lower MLD.

Figure R15. Structural characteristics of dsRNA. (a-c) Length distributions of dsRNA (L_{dsRNA}) under varying salt concentrations. (d) Scalar order parameter S estimated at different salt concentrations. (e) MLD of pgRNA as a function of salt concentration.

Corresponding changes:

1. (Supplementary Material) We have added Fig. S22 in this revised manuscript.
2. (page 12, paragraph 294) The following sentence was added:

“Meanwhile, the dsRNA becomes shorter as the salt concentration increases (Fig. S22a-c).”

3. (page 13, line 309) The following sentences were added to describe the dsRNA structural characteristics:

“To further characterize the structural organization of the pgRNA genome within the HBV capsid, we estimated the averaged maximum ladder distance (MLD) among the sampled structural ensemble of pgRNA in CG simulations, following previous works.^{73, 74} The average MLD value ranges between 110 and 120 under conditions where phase separation is prominent (Fig. S22d). This value is significantly lower than the MLD of

a randomly permuted single-stranded RNA of similar sequence length (~300 for a 3000-nt RNA), consistent with previous findings that viral RNAs are more compact than their randomized counterparts.^{74, 75.}

4. (page 12, line 296) *The following sentences were added in this revised manuscript:*

“To confirm this finding, we calculated the scalar order parameter S , defined as the largest eigenvalue of the local Q -tensor. Specifically, S was evaluated for each spatially adjacent pair of dsRNA segments and then averaged across all pairs. Across different salt concentrations, the average value of S ranged between 0.3 and 0.5 (Fig. 6i), which is characteristic of systems in the nematic phase.⁷² As expected, the average S value decreased with increasing salt concentration, reflecting the enhanced flexibility of dsRNA under high-salt conditions.”

5. Viruses tend to be overcharged, that is, there is more negative charge on the RNA than needed to compensate for the positive charges on the ARMs in the virus. This causes a Donnan equilibrium to establish itself for the mobile ions inside and outside the virus, and between the empty core of the lumen of the capsid and the region where ARMs and RNA interact. This equilibrium takes time to set up, and I wonder if the all-atom simulations are sufficiently long to let this equilibrium establish itself. Also, the Debye-Hückel potential used by the authors for the coarse-grained simulations most certainly cannot deal with this kind of phenomenology. It would be good to see if for physiological conditions Donnan effects can indeed be neglected by calculating time dependent ion density distributions in the atomistic simulations.

Reply:

We thank the reviewer for the valuable comments and suggestion. By analyzing the ion concentrations obtained from the all-atom MD simulations, we found that the number of mobile ions flowing in and out of the capsid becomes comparable within 10ns (Fig. R14a). Moreover, time-dependent radial distributions of ion concentrations become converged within the simulation timescale (Fig. R14b,c). These results suggest that the Donnan equilibrium can be established within the timescale of the all-atom MD simulations. The converged radial distributions of ion concentrations further showed that the relative ratio of the Na^+ concentrations between the inner and outer sides of the capsid is ~2.3 (Fig. R14b), which is considerably smaller than the values typical observed in other dsDNA viruses (>10) (Cosic et al, Nature, 2024, 627: 905-104). One possible reason of this smaller Donnan equilibrium effect is that at the pgRNA intermediate stage of the HBV life cycle, only a single RNA strand is present. In this case, the positive charges of the CTD segments (+3600 in total) and negatively charged RNA (-3200 not including polyA tail) are approximately balanced. With the progression of reverse transcription of pgRNA into dsDNA, the negative charges of the genome will be roughly doubled and therefore will become much larger than the positive charges of the CTDs. Under such conditions, the Donnan equilibrium effect

would become more pronounced.

As commented by the reviewer, Debye-Hückel potential was used in the coarse-grained model to account for ion effects. Therefore, the Donnan equilibrium cannot be captured within such an implicit solvent model. While neglecting this effect in the coarse-grained model may limit the quantitative interpretation of the simulation results, the major finding of CTD-pgRNA condensation and its impact is expected to be less sensitive to this limitation, as the positive charges of CTDs and negative charges of pgRNA are roughly balanced.

In this revised manuscript, we have incorporated these additional results and discussions to illustrate the Donnan equilibrium effect.

Corresponding changes:

1. (Supplementary Material) Fig. S2 was added in the Supplementary Material of this revised manuscript.
2. (page 3, line 86) The following sentences were added to discuss the ion distribution and Donnan equilibrium effect:

“By analyzing ion concentrations from the all-atom MD simulations, we found that the number of mobile ions flowing in and out of the capsid becomes comparable within 10 ns (Fig. S2a). Moreover, time-dependent radial distributions of ion concentrations become converge within the simulation timescale (Fig. S2b,c). These results suggest that the Donnan equilibrium can be established within the timescale of the all-atom MD simulations. The converged radial distributions further demonstrate⁹⁰ that the distribution of mobile ions inside the HBV capsid exhibits notable heterogeneity (Fig. S2d,e). In addition, the average concentration of Na⁺ inside and outside the capsid are 0.20 mM and 0.086 mM, respectively. The resulted relative ratio of the Na⁺ concentrations between the interior and exterior of the capsid (~2.3) is considerably smaller than the values reported for other dsDNA virus (>10)¹⁹. A likely explanation for this weaker Donnan equilibrium effect is that only a single RNA strand is present at the pgRNA intermediate stage of the HBV life cycle. In this case, the positive charges of the CTD segments (+3600 in total) and the negative charges of pgRNA (-3200) are approximately balanced. With the progression of reverse transcription of pgRNA into dsDNA, the negative charges of the genome will significantly exceed the positive charges of CTDs, which would lead to more pronounced Donnan equilibrium effect.”

3. (page 16, line 458) The following sentences were added in the Methods section of this revised manuscript:

“The Debye-Hückel type electrostatic potential was used to characterize the salt-concentration-dependent interactions of charged particles. It is worth mentioning that with such an implicit solvent model, the Donnan equilibrium cannot be explicitly

captured. While neglecting this effect in the coarse-grained model may limit the quantitative interpretation, the major results are expected to be less sensitive to this limitation as the positive charges of CTDs and negative charges of pgRNA are roughly balanced.”

Reviewer #3 (Remarks to the Author):

Bian et al computationally show that interaction of RNA with the basic C terminal domain of the HBV core protein results in a condensed phase. The condensate resembles the hollow shell of density seen in image reconstructions of HBV with the pregenome and with random RNA from the expression system. In the condensed phase the RNA is able to diffuse. As one would expect, neutralisation of phosphates allows base pairing. Unsurprisingly, the dsRNA segments shows nematic organisation. Unsurprisingly, the heterogeneity of the layer follows the symmetry of the capsid. To support their in silico studies, the authors demonstrated that isolated CTDs were able to form a phase when mixed with 15 nucleotide poly A oligomers.

There are several issues the authors need to address.

1) The approach to adding pgRNA to a capsid is novel; do all trajectories lead to the same result? How much do they differ?

Reply:

We thank the reviewer for the questions. For most cases, we carried out three independent simulations. The analysis confirms that simulations under identical conditions yield comparable phase behavior. To quantitatively assess variability across MD simulation trajectories, we incorporated error bars (estimated from the three replicates) into the distributions of key metrics, including the densities of the low- and high-density regions, the pgRNA mean squared displacement (MSD), and secondary structure content. The results demonstrate consistent outcomes across independent simulations under identical conditions. In this revised manuscript, we have also updated the figure captions to explicitly highlight these error bars, and the related discussion has been added accordingly.

Corresponding changes:

1. Following the reviewer’s comment on the reproducibility of the simulations, the estimation of error bars from three independent replicates was highlighted in the related figure captions.

2) While the peptide–RNA oligomer complex is experimentally shown to form a liquid-liquid phase separation, the two dimensional nature of the inner layer of the capsid

stretches the definition of a phase separation. This needs to be addressed.

Reply:

We thank the reviewer for the valuable comments and question. The key feature of the CTD-RNA phase separation along the inner surface of HBV capsid is that one end of the CTDs is anchored, effectively leading to a two-dimensional system. Although the CTD-RNA condensation of this two-dimensional system shares some important features with typical macroscopic LLPS in free solution, such as coexistence of high- and low-density regions, liquid-like dynamics of the condensates, and high sensitivity to perturbation in temperature and salt concentration, it has the following differences. First, the CTD is anchored to the inner surface of capsid, and therefore it cannot diffuse freely to form larger droplet by co-condensation with pgRNA. Second, the finite size of the system makes the condensate structure distinct from the round-shaped droplets typically observed in bulk-phase LLPS. To further characterize such two-dimensional phase separation for the CTD-RNA system, we performed atomic force microscope (AFM) characterization and observed morphological changes (Fig. R10). We further performed additional all-atom MD simulations for the two-dimensional CTD-RNA system and observed the segregation of CTD-RNA into high- and low-density regions, supporting the two-dimensional phase separation (Fig. R11). Following the reviewer's comments, we added the above results and discussion to address the differences of two-dimensional phase separation compared to the LLPS in free solute in this revised manuscript.

Corresponding changes:

1. (page 15, line 371) The following paragraph was added in the main text of this revised manuscript to address the features of two-dimensional phase separation:

“It is also worth emphasizing that the above discussed phase separation of the pgRNA genome within the HBV capsid differs from the macroscopic LLPS in free solution in the following two important aspects: i) the CTD is anchored to the inner surface of capsid and therefore cannot diffuse freely to form larger droplets through co-condensation with pgRNA; and ii) the finite size of the system makes the condensate structure distinct from bulk-phase LLPS, where low- and high-density phases are typically well separated to form round-shaped droplets. At the same time, this CTD-RNA co-condensation shares key features with typical macroscopic LLPS, including the coexistence of high- and low-density regions, liquid-like condensate dynamics, and sensitivity to temperature and salt concentrations.”

3) The phase needs to be better described. Do the high RNA density clusters show a characteristic arginine-RNA ratio? How fluid are the CTD-RNA clusters? How fluid are the dsRNA segments? If sufficiently stable, they will halt reverse transcription. A quantitative view of their stability will be appreciated.

Reply:

We thank the reviewer for the insightful comments and suggestion. In this revised manuscript, we performed a more thorough characterization of the condensate phase and analyzed the stability of dsRNA segments. The results reveal that within the dense phase, the arginine-RNA ratio stabilizes at a characteristic value of 1.05~1.11 across varying salt concentrations where phase separation is prominent (Fig. R16c). This value is lower than the overall system average among all the CTDs and pgRNA (~1.13), implying a fractionation effect that is typical of multicomponent phase separation systems. In addition, the CTD-RNA interactions are highly dynamic and flexible, as evidenced by substantial variations in the intermolecular contacts for a chosen pair of CTD and pgRNA segments (Fig. R16d). While the mobility of the CTDs themselves is constrained by their grafting to the inner capsid surface, the encapsulated RNA exhibits considerable fluidity, which is clearly demonstrated by the significant mean squared displacement (MSD, Fig. 2k). We also found that the dynamics are different in the ssRNA region and dsRNA region (Fig. R16e). Compared to the ssRNA region, the fluidity of the dsRNA region is much suppressed. In addition, dsRNA segments, including the typical protein binding motifs can dynamically fold and unfold dynamically (Fig. R1b-d), demonstrating the flexibility of dsRNA segments. We also observed a length dependence of the dsRNA stability as quantified by their lifetime (Figure R1f), which were estimated using a maximum-likelihood method.

Figure R16. Interaction modes between CTD and dsRNA from CG simulations of the pgRNA-containing HBV capsid. (a,b) A representative conformation showing the interactions between CTD (blue) and dsRNA (wheat) as viewed from the exterior (a) and interior (b) of HBV capsid, respectively. (c) Ratio of the numbers of arginine in CTDs and the nucleotides in RNA within high-density regions under different salt concentrations where phase separation is prominent. (d) The contact map between a randomly selected CTD segment and a specified pgRNA segment at two different simulation snapshots. The large change of the contact maps suggests the dynamic

nature of the base pairs. (e) MSD of dsRNA and ssRNA within a short lag time.

Corresponding changes:

1. (Supplementary Material) Fig. S20 was added in the Supplementary Material of this revised manuscript.

2. (page 13, line 316) The following sentences were added in the main text of this revised manuscript:

“Notably, the CTD-RNA interactions are also highly dynamic, as evidenced by substantial variations in the number of CTD-RNA contacts within the clusters (Fig. S20d). More detailed analysis revealed that the arginine-RNA ratio in the dense phase stabilizes at a characteristic value of approximately 1.05~1.11 across varying salt concentrations (Fig. S20c). This value is lower than the overall system average of 1.13, indicating a fractionation effect typical of multicomponent phase separation system⁵⁶

3. (page 12, line 263) The following sentences were added in the main text of this revised manuscript:

“The formed base-pairs exhibit pronounced dynamics in the CG simulations. Unwinding events of dsRNA can be clearly observed in the simulation trajectories (Fig. S17b and Supplementary Movie 11). Moreover, substantial variation in the number of formed dsRNA segments during the CG MD simulations supports the highly dynamic nature of the dsRNA conformation (Fig. S17e). The stability of dsRNA is length-dependent. Segments containing more base pairs exhibit longer lifetimes (Fig. S17f), which were estimated using a maximum-likelihood method. While the mobility of the CTDs is constrained by their anchoring to the inner capsid surface, the encapsulated RNA displays considerable fluidity, as evidenced by dynamic changes in local density at specified capsid site (Fig. 2j) and by its significant mean squared displacement (Fig. 2k). The dsRNA segments also exhibit pronounced mobility, although markedly reduced compared to the ssRNA region (Fig. S20e). This structural plasticity is functionally important for HBV pgRNA, as overly stable or rigid secondary structures could interfere with subsequent reverse transcription.”

4) One of the advantages of molecular dynamics, even with coarse graining, is the ability to visualize molecular contacts. How do the CTDs interact with the RNA? Do they interdigitate between strands? Do they fit in major grooves of dsRNA?

Reply:

We thank the reviewer for the suggestions. In this revised manuscript, we added visualization analysis to the results of molecular dynamics simulations. As shown in a representative region containing dsRNA, CTDs tend to be embedded between adjacent dsRNAs, acting as bridges that connect neighboring dsRNA strands (Fig.

R16a). Although we can also observe that CTDs fit to the major or minor grooves of dsRNA, the occurrences of such specific interaction modes are not dominant.

Corresponding changes:

1. (Supplementary Material) Fig. S20 was added in the Supplementary Material of this revised manuscript.
2. (page 13, line 314) The following sentences were added in the main text of this revised manuscript:

“We also analyzed the interaction characteristics between RNA and CTD. Structural visualization showed that CTDs tend to be embedded between adjacent dsRNA segments, acting as bridges (Fig. S20a,b). Although CTDs can also fit into the major or minor grooves of dsRNA, such specific interaction modes are not dominant.”

5) The authors appear to use the HBV pgRNA in their study but do not identify the sequence used. It is not clear if they have a polyA tail.

Reply:

We thank the reviewer for the comments. In our study, the RNA sequence is a wild-type sequence without the polyA tail derived from the complete genome sequence (GenBank accession number V00866.1). The 5'- and 3'-termini of the pgRNA were determined by referring to the information documented in a previous study (*J. Mol. Biol.* 2007, 370: 471–480). Following the reviewer's comments, we have added the description of the HBV pgRNA sequence to improve clarity.

Corresponding changes:

1. (page 16, line 418) The following sentences were added to this revised manuscript:

“The initial structure of the 3200-nt pgRNA was generated using the Nucleic Acids Builder (NAB) script in Ambertools¹⁰¹ with the wild-type pgRNA sequence. The pgRNA sequence without the polyA tail was retrieved from GenBank (accession number V00866.1). The 5'- and 3'- termini of the pgRNA sequence were identified by referring to the information documented in a previous study¹⁰².”

6) The means of adjusting ionic strength are not clearly defined. Are the authors holding the capsid interior to the stated bulk ionic strength? is the exterior held at the stated ionic strength? If the latter, what is the interior ionic strength or total number of ions?

Reply:

We thank the reviewer for the comments. In the all-atom MD simulations, the average ion strength is fixed. The mobile ions inside the capsid are freely exchangeable with those of exterior environment through the capsid cavities, leading to uneven ion distribution between the inside and outside of capsid. To quantify the spatial distribution of ionic strength, we calculated the radial distributions of Na⁺ and Cl⁻ concentrations (Fig. R14). The results revealed a highly inhomogeneous distribution of the ionic strength. The average concentration of Na⁺ (Cl⁻) inside and outside the capsid are 0.20 mM/mL and 0.086 mM/mL, respectively. In this revised manuscript, we added the above results and discussion to address the spatial distribution feature of ionic strength.

Corresponding changes:

1. (Supplementary Material) Fig. S2 was added in the Supplementary Material of this revised manuscript.

2. (page 16, line 429) The following sentence was added in the Methods section of this revised manuscript:

“Here the salt concentration corresponds to the average salt concentration of the whole system.”

3. (page 5, line 89) The following sentence was added in the Methods section of this revised manuscript:

“The converged radial distributions further demonstrate that the distribution of mobile ions inside the HBV capsid exhibits notable heterogeneity (Fig. S2d,e). In addition, the average concentration of Na⁺ inside and outside the capsid are 0.20 mM and 0.086 mM, respectively.”

7) There are several papers that are germane to this study that ought to be discussed to provide deeper context. Most importantly, Harati Taji et al (DOI: 10.1021/jacs.1c12439) showed that HBV-RNA interactions were highly fluid in NMR studies. How does your study conform to Harati Taraji predictions? Chu et al (DOI: 10.1128/JVI.03235-13) showed that HBV CTDs could act as RNA chaperones. Is this consistent with your results? Do dsRNA duplexes readily break in your simulations? Patel et al (DOI: 10.1038/nmicrobiol.2017.98) identify putative packing sites; are they folded in your study and how would a stable fold affect your predictions. Abraham and Lob (DOI: 10.1128/JVI.80.9.4380-4387.2006) predict long-range base pairing; how would such base pairing bias the packaging of RNA.

Reply:

We thank the reviewer for bring these important studies into our attention. As commented by the reviewer, these previous papers are highly relevant to the findings

of this study. For example, the highly fluid feature of the HBV-RNA interactions revealed by Taji et al by NMR studies is in line with the liquid-like dynamics of the CTD-RNA condensate formed by LLPS as discussed in this work. The observed role of LLPS in regulating the pgRNA base pairing support the chaperone activity of ARMs suggested in previous work by Chu et al. Interestingly, we can also observe the breaking events of dsRNA duplexes in the coarse-grained MD simulations. Moreover, as highlighted by Patel et al., RNA secondary structures, particularly those involving packaging signals, are important for HBV virion assembly. Here, we find that secondary structures formed by local sequence segments (e.g., the cis-acting element ϵ , *capsid protein packaging site PS3*) frequently form and break during simulations. This dynamic behavior appears functionally important, as excessively stable or rigid structures could interfere with subsequent reverse transcription. The biological relevance of the phase separation enhanced formation of long-range base pairing can also be supported by the study of Abraham and coworkers on the role of long-range base pairs for template switching involved in the reverse transcription. In this revised manuscript, we have refined the discussion to better highlight the connections between these previous studies and our current work, providing a more comprehensive context for the present study.

Corresponding changes:

1. (page 7, line 195) The following sentences were added in this revised manuscript:

“Interestingly, the highly fluid feature of HBV-RNA interactions within the capsid has been observed by Taji et al through nuclear magnetic resonance (NMR) studies,⁶⁰ consistent with the liquid-like dynamics of the CTD-RNA condensate described in this work.”

2. (page 12, line 296) The following sentences were added in this revised manuscript:

“Interestingly, the arginine-rich CTD has been suggested to act as a specific nucleic acid chaperone to facilitate the structural organization of the viral genome.^{29, 71} The CTD-induced pgRNA condensation and base-pairing observed in this study is in line with the chaperone role of CTDs as proposed in previous studies.

3. (page 12, line 270) The following sentences were added in this revised manuscript:

“This structural plasticity is functionally important for HBV pgRNA, as overly stable or rigid secondary structures could interfere with subsequent reverse transcription. Furthermore, the formation of secondary structures through local base-pairing in pgRNA, particularly sites involving packaging signals, has been proposed to play a critical role in the assembly of the HBV capsid.⁶⁷⁻⁷⁰ Notably, Patel et al. demonstrated that folding of packaging sites into stem-loop structures promotes capsid assembly⁶⁷. In line with this, our CG simulations showed that the segment corresponding to a

potential packaging site (PS3) can spontaneously form stem-loop-like structure (Fig. S17d). In addition, another important protein recognition motif, the cis-acting element ϵ , can also fold to stem-loop-like structure, although it involves frequent folding and unfolding transitions (Fig. S17c)."

4. (page 13, line 334) The following sentences were added in this revised manuscript:

"Notably, Abraham et al. demonstrated that interactions between the cis-acting element ϵ and the distal φ site facilitate template switching from ϵ to DR1,⁶⁹ highlighting the importance of long-range base pairing in reverse transcription. Here, we can observe an increase of long-range base pairing (Fig. 7d and Fig. S23), which is essential for enabling the template switches as discussed above. In this study, the base pairs with sequence separation larger than 100 are classified as long-range base pairs."

Responses to Reviewer's Questions and list of changes

Reviewer #1 (Remarks to the Author):

The authors have done an incredible job with addressing all comments from the previous round of review. I recommend publication in its present form.

Reply:

We thank the reviewer for the kind comments and for recommending its publication.

Reviewer #2 (Remarks to the Author):

I am satisfied with the extensive responses of the authors to the queries and comments of all three reviewers, and with the substantial revision of the manuscript that includes a discussion of additional studies that makes it an outstanding one.

Reply:

We thank the reviewer for the kind comments and the positive assessment.

Reviewer #3 (Remarks to the Author):

This revised manuscript addresses this reviewer's questions. With this revised ms, I better understand the effect of phase separation on polymerase location, the molecular interaction between CTDs and RNA, and the effect of CTDs on RNA mobility. The agreement between computation and experimental observation is convincing, notably the asymmetric reconstruction in reference 66, and will lead to a far better understanding of the process of in-capsid reverse transcription peculiar to HBV. This mechanism of RNA packaging is also likely relevant to many other small RNA viruses.

Reply:

We thank the reviewer for the careful review and kind comments on this manuscript.

There are several minor typos and clarifications to address:

15 "rich in" instead of "rich of"

40 some HBV serotypes have a 183 residue length

75 notably

88 delete "become"

91 are the Na⁺ concentrations in units of M or mM?

221 "AFM morphology" or "atomic force microscopy (AFM)"

235, 237, etc there are no sixfold axes. There are quasi-sixfold vertices

298 define Q-tensor

304 provide units for "100". Is it Å, bp?

312 provide units for ladder length (Å or bp)

337 units

Reply and corresponding changes:

We thank the reviewer for suggesting the changes and for pointing out the typos. In this revised manuscript, we made the suggested modifications and corrected the related typos.

Reviewer #4 (Remarks to the Author):

Reply:

We thank the reviewer for the carefully reviewing of this manuscript.